

# Comparison of the fast Lyman-Alpha and LICOR hygrometers for measuring airborne turbulent fluctuations

Astrid Lampert[1], Jörg Hartmann[2], Falk Pätzold[1], Lennart Lobitz[1], Peter Hecker[1], Katrin Kohnert[3], Eric Larmanou[3,4], Andrei Serafimovich[3], and Torsten Sachs[3]

[1]Institute of Flight Guidance, TU Braunschweig, Hermann-Blenk-Str. 27, 38108 Braunschweig, Germany
[2]Alfred Wegener Institute for Polar and Marine Research, Bussestr. 24, 27570 Bremerhaven, Germany
[3]GFZ German Research Centre for Geosciences, Telegrafenberg, 14473 Potsdam, Germany
[4]Swedish University of Agricultural Sciences, Umeå, Sweden

*Correspondence to:* Astrid Lampert (Astrid.Lampert@tu-braunschweig.de)

**Abstract.** The properties of fast hygrometers, the Lyman-Alpha and different LICOR humidity sensors, are analysed in direct intercomparison flights on different airborne platforms. One vibration isolated closed-path and two non-isolated open path LICOR sensors were installed on the twin engine turbo-prop aircraft Dornier 128. The closed-path sensor provided absolute values and fluctuations of the water vapour mixing ratio in good agreement with the Lyman-Alpha. The signals of the two open-path sensors showed considerable high frequency noise, and the absolute value of the mixing ratio was observed to drift with time in this vibrational environment.

On the helicopter-towed sonde Helipod with very low vibration level the open-path LICOR sensor agreed very well with the Lyman-Alpha over the entire frequency range up to 3 Hz.

The results show that the LICOR sensors are well suited for airborne measurements of humidity fluctuations, provided that a vibrationless environment is given, and this turns out to be more important than close sensor spacing.

## 1 Introduction

Water vapour and clouds in the atmosphere have a large impact on the energy balance (Ramanathan et al., 1989), the hydrologic cycle (e.g. Chahine, 1992) and therefore on local and global climate (Trenberth et al., 2007; Zhou et al., 2011). A general increase in surface air moisture and humidity within the troposphere has been reported (IPCC, 2013). The global distribution of moisture is difficult to measure and model accurately due to its large spatial and temporal variability (e.g. Klaus et al., 2012). Satellite retrievals of the vertical water vapour distribution provide limited spatial resolution, e.g. 300 m in the vertical and 30 km in horizontal direction (Bender et al., 2011). For the quantification of atmospheric processes on local to regional scales, airborne measurements are required to fill the gap between large-scale, low resolution information from satellites and point measurements with higher vertical and temporal resolution, but limited in horizontal extent.

Also measuring humidity *in situ* with high accuracy is challenging. In the troposphere, the water vapour concentration in ppm varies over two orders of magnitude (e.g. Schneider et al., 2010). Even for well controlled conditions in a cloud chamber, intercomparison measurements of different hygrometers probing the same air simultaneously revealed discrepancies between





different measurement systems of around 10 %. For cold and dry conditions, as encountered in the upper troposphere and lower stratosphere, the instruments had a variation around the reference value of 20 % (Fahey et al., 2014).

For quantifying moisture transport from the surface into the atmosphere, a commonly deployed method is the eddy covariance technique. The turbulent fluxes of latent heat are calculated under the assumption that vertical transport takes place via turbu-

lent eddies. For calculating the latent heat flux, accurate meaurements of the fast fluctuations of the vertical component of wind speed and humidity are necessary.

Airborne sensors for meteorological research have to fullfil specific requirements. On the one hand, a high temporal resolution is needed in order to obtain a high spatial resolution for the moving platforms. On the other hand, long-term stability and high accuracy, if possible without the need of calibration, are essential. In practice, this leads to the combination of complementary

sensors for both high resolution and long-term accuracy.

In the following, the measurement principles of the different humidity sensors used in this study are shortly summarized.

## 1.1   Material absorption

Capacitive sensors are based on taking up humidity in a porous or hygroscopic material, changing the dielectric properties. An

example are the Vaisala Humicap sensors, which are used as the standard for radiosondes. However, the response times for increasing and decreasing humidity may differ significantly due to the different diffusion coefficient into the material and out of the material, and the temporal resolution is limited. For temperatures exceeding $0\,°C$, sufficient humidity, and with the help of extensive postprocessing or modelling, the relatively slow polymer-based absorption hygrometers are sometimes used for retrieving humidity fluctuations (Wildmann et al., 2014).

The typical calibration procedure consists of applying saturated salt solutions with different known relative humidities, and recording the sensor output, thus creating a calibration curve. The sensor is sensitive to contamination by e.g. sea salt, which may alter the sensor properties significantly, and make a regular cleaning and re-calibration necessary.

## 1.2   Atomic absorption

Sensors based on atomic absorption provide the advantage of a very fast response time with measurement frequencies exceeding $100\,Hz$, a sharp absorption line compared to the absorption bands of molecules, and a high degree of absorption, which requires only measurement cells of few mm compared to several cm for molecular absorption. The sensor is based on the emission of ultraviolet (UV) radiation with a wavelength of $121.56\,nm$ (transition of an electron from the first excited state n=2 to the ground state n=1 in the hydrogen atom, called Lyman-Alpha emission line). The required length of the measurement

chamber is only a few mm (Buck, 1973). As the Lyman-Alpha wavelength is strongly absorbed by water vapour, the signal in the ion chamber detector is weakened accordingly. The relation is given by the attenuation law of Lambert-Beer. The absorption of the Lyman-Alpha line has cross sensitivities with oxygen and ozone molecules. The about 100 times absorption by oxygen atoms can be corrected by taking into account pressure and temperature, as the fractional density is constant. The correction of





the absorption by ozone molecules (same order of magnitude) is only necessary in the stratosphere (Buck, 1976).

Calibration is done by applying air of known humidities and recording the detector signal as a calibration curve. The aging of the lamp, or degradation of the magnesium fluoride windows, lead to a reduced signal strength with time, which is interpreted as higher absorption, thus higher water content. This long-term drift makes a regular calibration of either the sensor or the

retrieved data necessary.

The Lyman-Alpha sensor has been operated as the standard fast humidity sensor in many research aircraft for several decades (Busen and Buck, 1995; Corsmeier et al., 2001; Drüe and Heinemann, 2007; Twohy et al., 1997). However, with the end of the life time of the radiation sources (glow discharge lamps) and difficulties in replacing them, a variety of other fast humidity sensors is now available. A similar system is the Krypton hygrometer KH20 of Campbell Scientific, USA, which has a cross

sensitivity to oxygen as well and therefore has to be calibrated carefully (Foken and Falke, 2012). For the research aircraft of NCAR, a new Lyman-Alpha sensor was built, which showed promising first results (Beaton and Spowart, 2012).

## 1.3 Molecular absorption

Hygrometers based on molecular absorption can be sub-divided into systems based on a laser light source and the simpler

and cheaper technique of using a broadband light source combined with interference filters. Laser sources and fibre optics are now easily available due to advances and application in telecommunications. The disadvantage of molecular absorption is line broadening with pressure, resulting from the impact of other molecules. These measurement systems include the Picarro greenhouse gas analyser (Crosson, 2008) and the Los Gatos Fast Greenhouse Gas analyser, which measure the three most important greenhouse gases water vapour, carbon dioxide and methane simultaneously. Further, the information of the amount

of water vapour is needed for correction due to cross sensitivity. On the DLR HALO (High Altitude Long-Range Aircraft), an innovative spectroscopic sensor developed by the Physikalisch-Technische Bundesanstalt (PTB) Braunschweig is deployed (Buchholz et al., 2014). Compared to the LICOR sensor, this tunable diode laser hygrometer can be operated much faster (up to several kHz) and with a known accuracy, providing the most precise humidity values available to date (Buchholz et al., 2013, 2014, 2016). However, this hygrometer requires extensive post processing, and at least so far it is not possible to obtain

real-time humidity data. The spectroscopic sensors are experimental systems and not commercially available.

The LICOR sensors LI-7500, LI-7500A and LI-7200 for measuring fast humidity are based on the absorption of near infrared radiation. They have a longer measurement chamber of 12.5 cm compared to the few mm of the Lyman-Alpha. The LICOR sensors in different forms are used at automated field stations for research networks covering large temporal and spatial scales, and they are therefore well characterised concerning the transfer function of different components (Metzger et al., 2016). The

closed-path sensors requiring a gas sampling system can be affected by high-frequency attenuation, so inlets and tubes have to be dimensioned reasonably (Aubinet et al., 2016; Metzger et al., 2016).

Like for the Lyman-Alpha, the attenuation of the signal obeys the law of Lambert-Beer relating absorptance to the number density of the absorbing gas, taking into account pressure, and with some correction terms due to cross-sensitivities. The calibration procedure consists of applying air of well-defined humidity for two points, and adapting the calibration coefficients





accordingly. Zero signal is created by dry, carbon dioxide free gas. The second point can be applied e.g. with the LICOR dew point generator.

The LICOR humidity sensors have been used for airborne applications additionally to the Lyman-Alpha or to replace the Lyman-Alpha for many years (Beringer et al., 2011; Pillai et al., 2011; Hiller et al., 2014). The sensors have been implemented

in various configurations, some facing with the sensor head forward, e.g. on the helicopter borne sondes Helipod, and the Airborne Cloud Turbulence Observation System ACTOS (Siebert et al., 2013), some with the sensor head oriented vertical (French Piper Aztec research aircraft), some with an open housing (Siebert et al., 2013), some with a metal grid (Helipod), some with additional purging with synthetic air to keep the detector free of water vapour (Schmitgen et al., 2004).

The manufacturer warns in the manual that the sensor should not be applied with vibrations around 150 Hz and around the

harmonics (Licor, 2014). However, the impact on measurement results is not specified or even quantified. As the LICOR sensor is currently the fastest and cheapest water vapour sensor commercially available and small enough to be easily integrated into aircraft, its airborne applications will very likely increase.

Therefore the aims of this article are to show limitations and provide information for successful handling for the airborne use of LICOR humidity sensors, to quantitatively compare fluxes of latent heat obtained with the former "standard" Lyman-Alpha

and with the LICOR sensors for airborne applications, and to determine the required measurement frequency for humidity fluctuations to derive reliable latent heat fluxes.

## 2   Experimental setup

This section provides the background of the sensors used in the study, including a short overview about the LICOR sensor working principle, and a description of the airborne platforms and their sensor setup. For the flights, both a Lyman-Alpha and

at least one LICOR sensor were operated in parallel, and the latent heat fluxes derived with the different humidity sensors and the same wind vector measurements are compared directly.

### 2.1   LICOR sensors

The working principle of the LICOR sensor series for water vapour and carbon dioxide ($CO_2$) is the absorption of near in-

frared radiation by $H_2O$ and $CO_2$. The radiation source is a small lamp with broadband emissions in the near infrared (NIR) spectral range. The radiation is focussed on a filter wheel with bandpass filters of different wavelengths, rotating at a speed of 150 Hz. The wavelength of 2590 nm is absorbed by water vapour but not by carbon dioxide ($CO_2$), the wavelength of 4260 nm by $CO_2$, but not by water vapour, and the wavelength of 3950 nm serves as a reference, where neither $CO_2$ nor water vapour have absorption bands. The narrow band-pass filtered radiation then passes the measurement cell of 12.5 cm length, where the

ambient air either passes passively (open path sensor, LI-7500, and the newer LI-7500A) or is pumped through (closed path sensor, LI7200), and partly absorbs radiation depending on the number density of the absorbing water molecules, as well as for the $CO_2$ concentration. The detector on the other side of the measurement cell is a thermopile, additionally cooled by Peltier





elements. The system is described more in detail in Licor (2014). The data obtained originally at 150 Hz frequency is internally processed and finally provided at a maximum frequency of 20 Hz. The delay time for internal processing is specified for the LI-7500 as 185 ms, and for both the LI-7500A and LI-7200 as 123 ms.

## 2.2   Do128 instrumentation

The standard meteorological equipment of the research aircraft Do128 "D-IBUF" of the Institute of Flight Guidance, TU Braunschweig, consists of a five-hole probe and corresponding pressure transducers of Setra (static, dynamic and differential pressure), inertial navigation and global navigation satellite system (GNSS) for deriving the 3D wind vector, a slow, but highly accurate Rosemount DB102 temperature sensor, and a Rosemount EL102 sensor with a fast response time. The temperature sensors are mounted in a sophisticated Rosemount inlet to obtain directly the static air temperature, and can additionally be heated for flights through icing conditions. The humidity channel includes a dew point mirror TP 3-S of Meteolabor (not used for this study), a capacitive humidity sensor (Vaisala HMP233 Humicap) and a Lyman-Alpha optical sensor L-6 / HMS-2 of Buck Research. Further, a surface temperature sensor KT15 is included. Temperature and humidity sensors, as well as the five-hole probe are integrated into the nose boom. More details on the instrumentation can be found in Hankers (1989); Bange et al. (2002); Corsmeier et al. (2001, 2002). The Vaisala Humicap is calibrated regularly, and the combined signal of the slow Humicap and the fast Lyman-Alpha has been shown to agree well with other independent measurements of humidity (Sodemann et al., 2017).

For the humidity intercomparison flight, three different LICOR systems were available (open path LI-7500 with serial number 75H-0775, open path LI-7500A with serial number 75H-2287, and closed-path LI-7200 with serial number 72H-0584). The two open path sensors were covered by a sheet metal with holes to enforce turbulent mixing and avoid gradients of concentration in the measurement cell. The three LICOR sensors were installed in addition to the standard equipment at the following locations: on the nose boom (LI-7500A, in the following called Li1), in the cabin directly under the roof (LI-7200, called Li2), with an inlet sampling the air near the LI-7500 (called Li3) on the roof (see Fig. 1). All sensors were oriented along the aircraft. No purging of the sensors with nitrogen or dried air was applied.

The stainless steel inlet to the Li2 sensor had a length $l_1$ of 350 mm, an inner diameter of 9.6 mm, resulting in an area of $A_1$=7.24·$10^{-5}$ m$^2$. The pump provided a flow of Q=15 l min$^{-1}$=2.5·$10^{-4}$ m$^3$ s$^{-1}$. For the tube, the airspeed is therefore $v_1 = \frac{Q}{A_1}$=3.45 m s$^{-1}$. This results in a time delay $\Delta t_1 = \frac{l_1}{v_1} = 0.1$ s. Similar calculations are applied for the nylon tube of length $l_2 = 400$ mm and inner diameter of 8 mm, guiding the air from the inlet to the sensor, which results in an additional time delay of 0.03 s. Additionally, the time for exchanging the air of the measurement cell with an inner diameter of 25 mm and a length of 125 mm amounts to 0.12 s. Altogether, there is a delay time of 0.25 s caused by the sampling system for the Li2.

For the humidity intercomparison flight, vibration sensors of type M3555B04/03/02 and M356B18 of the company PCB Piezotronics Inc., US, were integrated along the x-axis (in flight direction along the aircraft and the sensors) and z-axis (upward directed) of the LICOR sensors. For the Li2, another vibration sensor in y direction (perpendicular to flight direction)





was available.

### 2.3 Helipod instrumentation

The data of a measurement flight with the "Helipod", a meteorological sonde towed by rope to a helicopter (e.g., Bange and
Roth, 1999; Bange et al., 2002; Martin and Bange, 2014), is analysed here. The Helipod sonde is equipped with different mete-
orological sensors: Humidity measurements are performed with a Lyman-Alpha sensor L6 of Buck Research, US, a capacitive
Vaisala Humicap HMP110, a dew point mirror 1011B of General Eastern, US (not used for this study), and the LI-7500 (same
sensor as used for the Do128 flight, called there Li3). Temperature is measured with a Pt100 by Rosemount, and a fine wire
by Dantec. A five-hole probe with the same differential pressure sensors as in the Do128 is integrated (D289 for differential
pressure, D270 for static pressure, Setra, US), as well as a GPS system with eight receivers for a full 3D attitude alignment
(GNATTI System of Geo++ GmbH, Germany) and IMU (LCR 88, LITEF, Germany). A Heimann KT15 sensor (Heimann,
Germany) records the surface temperature. Altitude information is provided by GPS, barometric pressure, and a radar altimeter
ERT180 (Thomson-CFS, France).

## 15   3   Flights and atmospheric conditions

### 3.1   Do128 flight on 23 October 2015

The measurement flight with the research aircraft Do128 "D-IBUF" on 23 October 2015 was conducted above different terrain
of the North German Plain, including areas dominated by forest, by agricultural grassland, and above open water North of the
East Frisian Islands. For the analysis, the flight was sub-divided into six straight legs D1-D6 of 10 km length above terrain that
was chosen as homogeneous as possible: D1 above forest at an altitude of 430 m above mean sea level, D2 above agricultural
land at an altitude of 430 m, D3 above agricultural land at 220 m altitude, D4 and D5 above open water at 100 m altitude, and
D6 above forests at 270 m altitude (Fig. 2). Apart from these short legs, a longer part of the flight was used for the sensor
comparison (grey shaded in Fig. 3), which fullfills the requirements of Lenschow et al. (1994) for the sampling length.
The flight took place under varying cloud conditions, mostly overcast with a cloud bottom at an altitude of around 1000 m.
Above land, a neutrally stratified boundary layer was observed up to an altitude of around 1000 m, with a strong increase of
potential temperature of 6 K in 1000 to 1200 m (Fig. 4). However, above the North Sea, the atmosphere was stably stratified.
Therefore, significant latent heat fluxes were only observed above land.

### 3.2   Synchronisation of the Do128 humidity sensors

Before calculating the humidity fluxes, the four fast humidity sensors (Lyman-Alpha, Li1, Li2, Li3) onboard the Do128 located
at different places were synchronised. As mentioned in Sect. 2.2, the Lyman-Alpha and the Li1 are located at the nose boom,





the Li3 at the cabin roof, and the inlet of the Li2 is close to the Li3 on the cabin roof. First the Lyman-Alpha was shifted in time against the vertical wind speed by maximizing the turbulent latent heat flux. The synchronisation was then done by retrieving the maximum correlation of the covariance of the mixing ratio fluctuations of the Lyman-Alpha and each of the LICOR sensors for varying temporal offsets. The time step providing the highest correlation was used as the delay time between the sensors.

The time signal of the Lyman-Alpha was used as reference. The part of the flight used for the synchronisation (shaded in grey in Fig.3) contains large variability in the signals (changes of altitude, and high fluctuations), facilitating to derive a high correlation of the signals. The time shift between the Lyman-Alpha and the Li1 sensor in the nose boom was 0.15 s. As stated in the LI7500A manual, an internal delay of 0.13 s is caused by data processing. The remaining small difference of 0.02 s may be explained by the sampling geometry, as the Li1 is covered by a sheet metal as explained in Sect. 2. The time shift between

the Lyman-Alpha and the Li3 sensor at the cabin roof was 0.3 s. Therefore, the delay time between the two open path sensors Li1 and Li3 amounted to 0.15 s. This can be well explained by the different internal delay times of 0.13 s for the Li1 and 0.185 s for the Li3, amounting to 0.05 s, plus additionally the distance $\delta s$ of 7 m between the two sensors (Li1 installed at the nose boom, Li3 at the cabin roof) and the true airspeed $v_{tas}$ of 70 m s$^{-1}$:

$$\Delta t = \frac{\Delta s}{v_{tas}} = \frac{7\,m}{70\,m\,s^{-1}} = 0.1\,s \tag{1}$$

The time shift between the Lyman-Alpha and the Li2 in the cabin was 0.45 s. Subtracting the values for the time shift caused by the distance (0.1 s) and the internal delay (0.13 s), this results in a time difference of 0.22 s. This delay can be explained by the tube length and flow speed for guiding the air to the measurement cell in the cabin, as calculated in Sect. 2.

For further calculations, the best fitting time shift correction was applied to all three LICOR sensors. The correlation between the LICOR signals and the Lyman-Alpha amounted to values exceeding 0.95 for the Li2 and Lyman-Alpha, and exceeding 0.8

for the Li1 and Lyman-Alpha for the part of the flight considered here. The best correlation between Li3 and Lyman-Alpha amounted to values between 0.5 and 0.9, thus was considerably lower.

The calculations of the best fitting time delay were verified by maximizing the coherence spectra of the Licor sensors and the Lyman-Alpha as a reference, and at the same time minimizing the phase between the two signals. This resulted in the same delay times as determined by the method of maximizing the correlation.

## 25 3.3 Helipod flight on 14 August 2014

The overall aim of the Helipod measurements was to study greenhouse gas emissions on a scale of up to 100 km to investigate the spatial variability, and to analyse how representative the continuous emission measurements on local scales are on a climatically relevant sub-regional scale. During the measurement flight, the Helipod was attached by a 30 m rope to a Russian Mi8 helicopter. The flight was performed from the Research Station Samoylov Island in the Lena Delta, Siberia.

The Helipod flight analysed here took place on 14 August 2014, when thawing was still in progress. The flight pattern consisted of a vertical profile up to 1500 m altitude, then a straight, horizontal flight leg of around 100 km at low altitude (around 100 m) towards North-West. There, another vertical profile was performed, and the flight back followed the same track. The atmosphere was neutrally stratified in the lowermost 150 m, then slightly stable up to 1000 m (increase in potential temperature





smaller than 1 K from the urface up to that altitude), and the ABL top was evident by an increase of potential temperature at 1000 m altitude (not shown).

The flight on 14 August 2014 was done in conditions nearly free of clouds at the beginning with a near-surface air temperature around 17 °C and southerly wind with a speed of $5\,\mathrm{m\,s^{-1}}$. For this instrumental intercomparison, a long flight transect from

Samoylov Station to Arga-Muora in straight North-Westerly direction at around 100 m altitude is analysed (Fig. 5). On the way back, short rain showers were encountered. The time series of the height, vertical wind speed, mixing ratio, potential temperature and valid data is shown in Fig.6.

### 3.4   Synchronisation of the Helipod humidity sensors

As the LI-7500 system was calibrated directly before the measurement campaign, and therefore provides reliable absolute

values, the Lyman-Alpha values of the mixing ratio were calibrated against the LICOR data using a linear regression method. Before calculating turbulent fluxes of latent heat, the time shift between the LICOR and the Lyman-Alpha was corrected by calculating the maximum coherence with minimum phase shift of the two signals. The best correlation was found for a total time shift of 0.315 s. This includes the time delay caused by internal processing of 186 ms, plus an additional time shift of around 130 ms. This value was confirmed by calculating the best correlation between the time series. The time lag was the

same for straight and level flight sections throughout three campaigns in Siberia in April, June and August 2014.

## 4   Results

### 4.1   Vibrations during the Do128 flight

The time series of the mixing ratio (Fig. 3) shows the general behaviour of the Li2 sensor, the slow Humicap and the fast Lyman-

Alpha sensor. The time periods affected by radio communication were excluded for further analyses, as they occasionally induce artificial spikes on the Lyman-Alpha sensor. The data used for calculating the spectra are shaded in grey. They were chosen to exclude the flights at higher altitude, where the signal of the Lyman-Alpha differs significantly from the other sensors. Under these different pressure conditions, a different sensitivity range would be necessary, which was not adapted during the flight. For the data of the Li1 and Li3, different effects can be observed:

– The signals of the open path sensors Li1 and Li3 contain a higher level of noise compared to the closed path Li2 system and the signal of the Lyman-Alpha.

    – Changes in altitude affect the signals of Li1 and Li3, and the absolute values of the mixing ratio do not follow the behaviour of the Li2 and the Humicap.

    – There is a general slow drift in the signal of the open path sensors Li1 and Li3, which does not follow the trend of the

closed path Li2 sensor, the signal of Lyman-Alpha, and Humicap.



Differences are apparent in the time series of the vibrations (Fig. 7): The amplitude of acceleration in z direction is around $50\,\mathrm{m\,s^{-2}}$ for the Li3, around $40\,\mathrm{m\,s^{-2}}$ for the Li1 and around $3\,\mathrm{m\,s^{-2}}$ for the Li2 sensor. The example shows flight section D1, but is comparable for all sections.

The acceleration spectra for all three sensors in x and z direction (along sensor and aircraft axis, and vertical direction) as well as y direction (perpendicular to sensor and flight direction) for the Li2 sensor are shown in Fig. 8 for flight section D3. Strong and sharp vibration peaks at distinguished frequencies were recorded at the locations of all sensors, as well as differences in the broadband features.

Generally, the acceleration values for high frequencies (exceeding $200\,\mathrm{Hz}$) are several orders of magnitude lower for the Li2 sensor compared to the Li1 and Li3 sensors. This feature gets more pronounced for frequencies exceeding $1000\,\mathrm{Hz}$. The strength of vibrations contained within individual peaks is more pronounced for the Li1 and Li3 sensor compared to the Li2 sensor. In Fig.8, the critical frequencies of $150\,\mathrm{Hz}$ and higher harmonics, as specified by the manufacturer, are indicated by vertical black lines. Especially around $450\,\mathrm{Hz}$ it can be seen that the vibration level of the Li3 is more than an order of magnitude higher than the vibration level of the Li1. This feature is observed during all flight legs analysed here, persistent throughout each flight leg. The high level of vibrations at the critical frequencies and potential impact on the internal signal processing could be an explanation for the different humidity spectra shown in the following.

## 4.2 Powerspectra of humidity fluctuations

The above mentioned properties of the sensors are reflected in the spectra of the mixing ratio, shown in Fig. 9 for each humidity signal. The sloped lines represent the -5/3 drop-off expected in the inertial subrange of decaying turbulence. Between 0.003 and $0.3\,\mathrm{Hz}$ all sensors quite nicely follow the Kolmogorov prediction (Kolmogorov, 1941), and those of the Lyman-Alpha and Li2 continue for a further decade, while Li1 and Li3 level off indicating a substantial level of white noise superimposed on the humidity signal. The Humicap spectrum gradually decreases slightly faster.

Overall, the Lyman-Alpha spectrum most closely behaves as expected from the theory, and that for the Li2 sensor is very similar but drops off marginally faster beyond $1\,\mathrm{Hz}$. This behaviour cn be attributed to somewhat increased dampening due to longer inlet tubes in comparison with those for the Lyman-Alpha.

At low frequencies the Vaisala Humicap and even more the Li3 sonsor show higher variances. It will be shown in the next subsection that this variance is differently correlated with the vertical wind velocity, which has implications for the flux calculation.

## 4.3 Cospectral analysis

As the Lyman-Alpha humidity sensor has been used widely in turbulence studies over decades, it is taken as a reference to compare the behaviour of the other sensors. Cospectra between each of these other sensors and the Lyman-Alpha are calculated. Fig.10 (left column) shows the coherence and the phase for the flight sections marked grey in Fig.3. The overall best coherence with Lyman-Alpha shows the Li2, it is virtually equal to one with no phase difference over three decades, and only drops



off beyond 1 Hz as a result of the separation of the sensors. In the phase spectrum the coherence is coded in the thickness of the dots as a phase can only be interpreted if a significant coherence between the signals is present. The marginal positive phase between Li2 and Lyman-Alpha is a result of the linear advancement over 0.315 s of the Li2 signal. This linear shift can only approximate the more complex difference in the high-frequency response behaviour between Li2 and Lyman-Alpha

due to spacing and tubing. A smaller advancement, however, leads to reduced coherence and a trailing phase shift of Li2 for frequencies below 1 Hz. The other two LICOR sensors (Li1 and Li3) have far less coherence with the Lyman-Alpha. At low frequencies this reflects the drift of both vibration affected sensors, and at high frequencies the noise fades more coherence that might exist. Note that over all three LICOR sensors the coherence inversely correlates with the amount of vibration the sensors are exposed to. More complex is the reponse behaviour of the Vaisala Humicap. At low frequencies (<0.01 Hz) it agrees

reasonably with the Lyman-Alpha, then starts to trail it with increasing frequencies, but at e.g. 0.4 Hz it responds to some degree in phase with the Lyman-Alpha, but at a reduced level of coherence. To assess the sensor behaviour on the moisture flux calculation, Figure 10 (right) shows the covariance spectra of the vertical wind speed and the different humidity sensors after correction of the time lag. The spectral estimates are multiplied by the frequency, thus the area below the curves is proportional to the humidity flux. Flux estimates based on Li2 and Lyman-Alpha reasonably agree, but those calculated by the vibration

affected LICOR sensors are too low, most pronounced for Li3. The Humicap shows an interesting behaviour: overestimation on a scale of minutes (0.02 Hz) and underestimation for higher frequencies, both of which compensate to a certain degree. This behaviour seems to be a specific property of the Vaisala Humicap sensor.

Finally the total moisture flux as calculated by the five different moisture signals was compared. In Fig.11 the integrated covariance spectra (ogives, see e.g. Sievers et al., 2015) are shown, normalised by the integral of the cospectrum of $w$ and

the Lyman-Alpha humidity. With the closed-path Li2, 98.1% of the Lyman-Alpha value is reached, the small high frequency loss is due to different sensor spacing and tubing. The vibration affected open-path LICOR sensors reach 47% and 83% of the Lyman-Alpha value.

It can be concluded that the sampling frequency of 20 Hz is sufficient for airborne turbulent humidity fluxes.

The scale of eddies corresponding to the frequency of 1 Hz and the airspeed of 70 m s$^{-1}$ is around 70 m. Contributions from

eddies smaller than the size of few 10 m are negligible. This information is visible in the cospectra and in the ogive functions. The results emphasise on which scales turbulent transport of humidity takes place: Fluctuations in the frequency range higher than 2 Hz do not contribute significantly to the overall humidity fluxes for an air speed of 70 m s$^{-1}$. This is different to flux measurements near ground, where high resolution sampling and close sensor spacing are essential (Caughey and Palmer, 1979; Kaimal and Finnigan, 1994; Bange et al., 2002).

## 4.4    Spectral analysis of mixing ratio for the Helipod flight

Fig.12 (left) shows the spectra of coherence and phase of the LI7500 signal (time corrected and not time corrected) against the Lyman-Alpha. For a frequency up to 3 Hz, the coherence for the time corrected signal is higher than 0.8, and the phase shift around 0 °C, thus the agreement of the two signals is high. The plot on the right represents the humidity flux for both sensors, i.e. the covariance spectra of the humidity and the vertical wind speed component. The areas under the curves are proportional





to the humidity fluxes. The Lyman-Alpha and LICOR signals agree perfectly in the frequency range up to 2 Hz. For higher frequencies, the humidity flux is negligible anyway. The effect of the time correction for the humidity fluxes can be seen for frequencies exceeding 0.2 Hz: There are differences in the area under the lines representing the time corrected and uncorrected values. The overall effect is in the range of few percent. As the LI7500 used on the Helipod flew on the Do128 as well (Li3),

the excellent agreement with the Lyman-Alpha here demonstrates that the vibrations of the Do128 are the main reason for poor performance there.

## 5   Recommendations for airborne applications

For calculating turbulent fluxes, the best temporal correlation of the sensors has to be determined first. The time shifts that were determined are not negligible and have to be taken into account. This is a standard procedure in the flux community (Moore,

1986). Time delays for the LICOR sensors are partly caused by internal processing, partly by different locations of the sensors and tube lengths.

For the Do128 application, the humidity fluxes determined by the three different LICOR sensors did not always match satisfactorily the humidity fluxes determined with the Lyman-Alpha sensor: For the Li2 sensor, which was installed isolated against vibrations, the latent heat fluxes were comparable to the Lyman-Alpha. However, for the Li1 and Li3 sensors, installed without

particular isolation agains vibrations, the correlation with the Lyman-Alpha signal was significantly lower, and the covariance spectra were different, resulting in larger deviations of the latent heat fluxes. As an explanation, the different levels of vibrations for the LICOR sensors at the specific location within the aircraft was shown. However, the spectral behaviour of the vibrations had no direct, linear impact on the humidity spectra, but the relationship is more complex. This is currently subject to more detailed investigations.

For the Helipod application with lower vibrations, after careful sensor calibration to absolute values and correction of the time lag, the humidity fluxes derived from the Lyman-Alpha and LICOR sensors agreed very well.

In summary, some precautions have to be taken for employing a LICOR sensor for airborne turbulent humidity flux measurements. Especially the level of vibrations and its impact on the measurements should be evaluated critically, and the spectra of the measurements should be checked for plausibility. Especially for small fluxes, the relative error might be significant.

Generally a temporal resolution of 20 Hz is sufficient for humidity flux calculations, as the contribution of the higher frequency parts of the spectrum is negligible.

*Competing interests.* The authors declare that they have no conflict of interest.




*Author contributions.* A.L. wrote the paper, F.P., J.H., A.L. and L.L. performed the data processing and analysis for Helipod and the Do128 flight. P.H. contributed to the analysis. A.L., F.P. and J.H. planned and conducted the Do128 flight with different LICOR humidity sensors. T.S., E.L., K.K. and A.S. planned and conducted the Helipod measurements. All authors commented and contributed to the manuscript.

*Acknowledgements.* The authors would like to thank Rolf Hankers, Mark Bitter and Helmut Schulz for support with the Do128 intercom-
5   parison flight. The authors thank the Alfred Wegener Institute, especially Martin Gehrmann, for providing the LI7200 for the flight, and
for fruitful discussions about sensor integration. The authors would like to thank Matthias Cremer from Messwerk GmbH for providing the
3D accelerometer for the Do128 flight. The authors wish to acknowledge all those who supported the field measurements during the Lena
Delta Experiment in 2014. The Helipod flight campaign was supported by the Helmholtz Association of German Research Centres through
a Helmholtz Young Investigators Group grant to T.S. (grant VH-NG-821) and is a contribution to the Helmholtz Climate Initiative REKLIM
10   (Regional Climate Change).





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



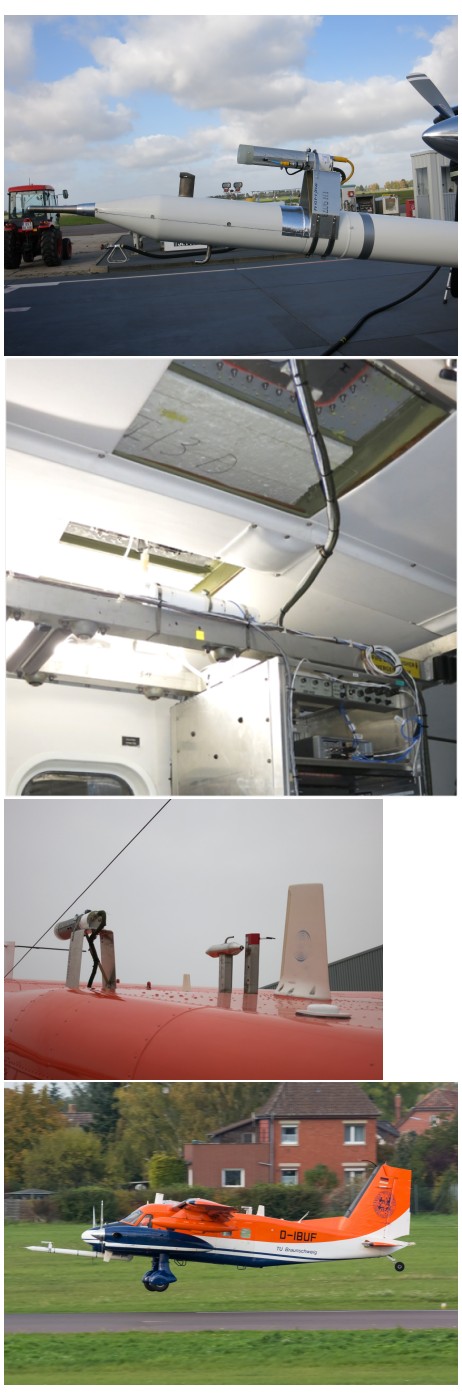

**Figure 1.** The three LICOR sensors integrated into the Do128 during a humidity intercomparison flight: a) LI7500A on the nose boom (Li1), b) LI7200 in the cabin with an inlet near the LICOR sensor on the roof (Li2), c) LI7500 on the roof (Li3), and d) the Do128 equipped with additional sensors during the flight (last photo courtesy of Uwe Bethke).



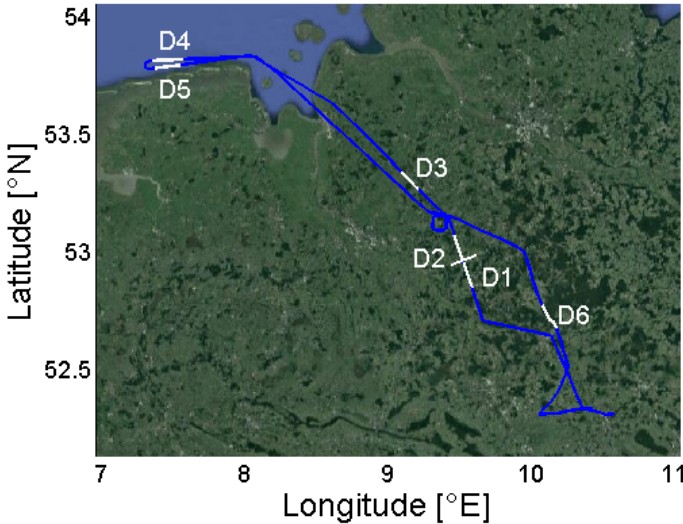

**Figure 2.** Flight track of the Do128 aircraft above the North German Plain during the humidity intercomparison flight. The flight sections D1 to D6 referred to in this study are indicated in white. The picture was taken from Google Maps, accessed on 29 May 2017.

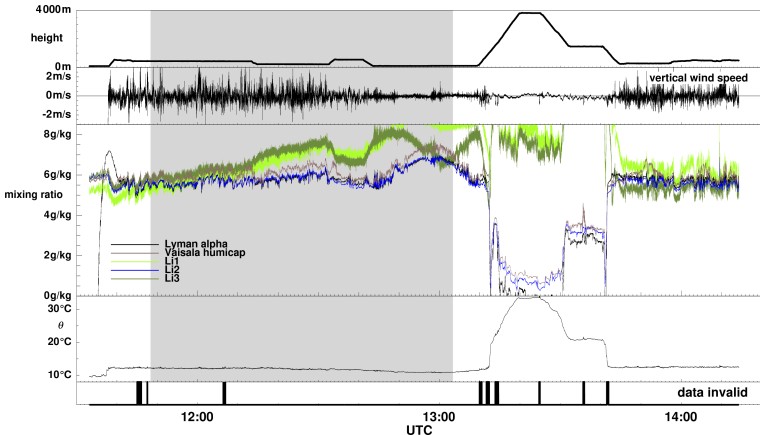

**Figure 3.** Overview of the flight with the Do128 on 23 October 2015. The first subplot shows the height, the second the vertical wind speed, indicating the strength of atmospheric turbulence. The main plot shows the time series of the mixing ratio measured by Lyman-Alpha (black), Vaisala Humicap (purple), closed-path Li2 (blue), open-path Li1 at the nose boom (bright green), and closed-path Li3 on the cabin roof (olive). Then there is the time series of the potential temperature. The last subplot shows the invalid data, e.g. when radio communication disturbed the signals. For the spectral analysis, the part of the data shaded in grey were used, excluding segments with invalid data, e.g. disturbance by radio communication.




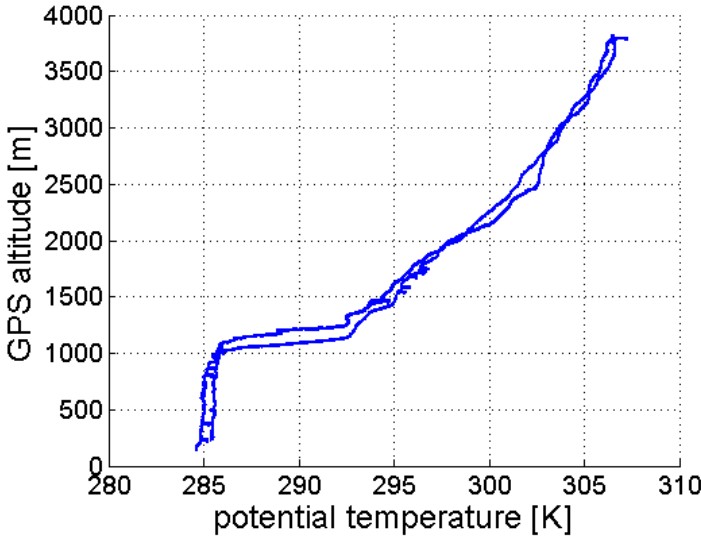

**Figure 4.** Vertical profile of the potential temperature recorded on 23 October 2013 during the flight with the Do128. The profiles were recorded on the way back above land, North-West of leg D3.

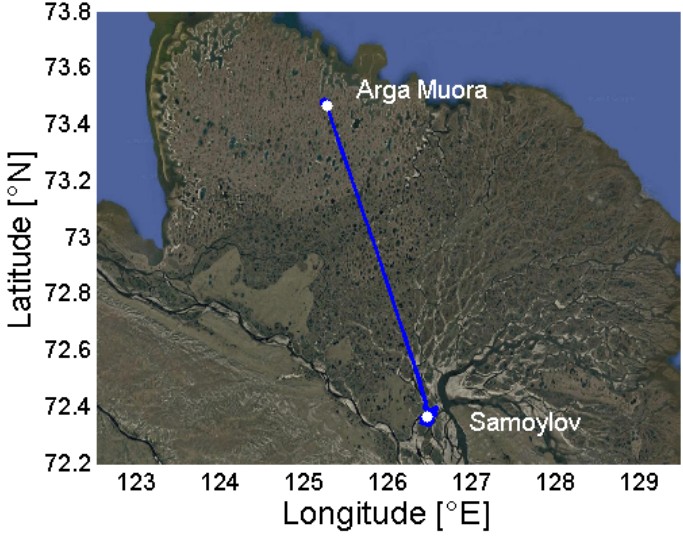

**Figure 5.** Flight track of the Helipod in the Lena River Delta for the measurement flight on 14 August 2014. The flight started at Samoylov Station and went towards Arga-Muora in the North-West. The picture was taken from Google Maps, accessed on 29 May 2017.




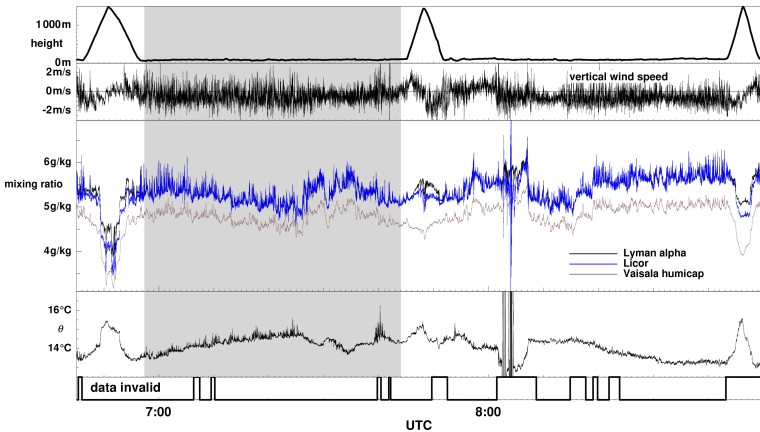

**Figure 6.** Overview of the flight with the Helipod on 14 August 2014. The first subplot shows the height, the second the vertical wind speed, indicating the strength of atmospheric turbulence. The main plot shows the time series of the mixing ratio measured by Lyman-Alpha (black), Vaisala Humicap (purple) and the open-path Li3 (blue). Then there is the time series of the potential temperature. The last subplot shows the parts of the flight with data that was excluded from the analysis for various reasons (e.g. impact of rain). For the spectral analysis, the part of the data shaded in grey were used, excluding segments indicated in the last plot.

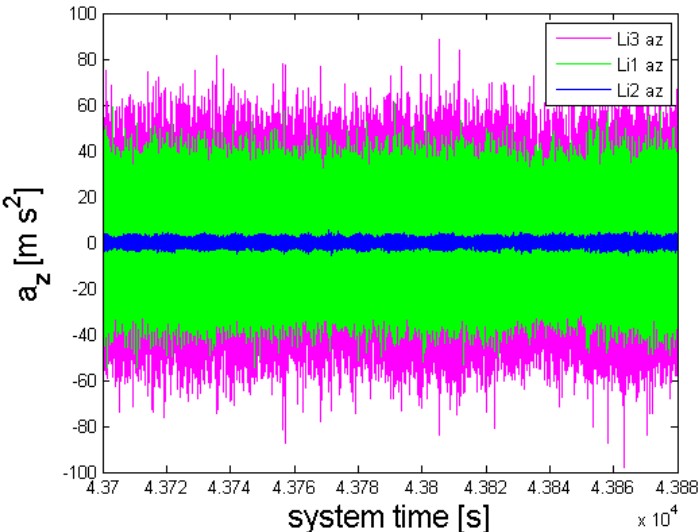

**Figure 7.** Time series of the vibrations in z direction of the three LICOR sensors during the flight section D1 on 23 October 2013. The vibration measurements at the Li1 are shown in green, at the Li2 in blue, and at the Li3 in magenta.





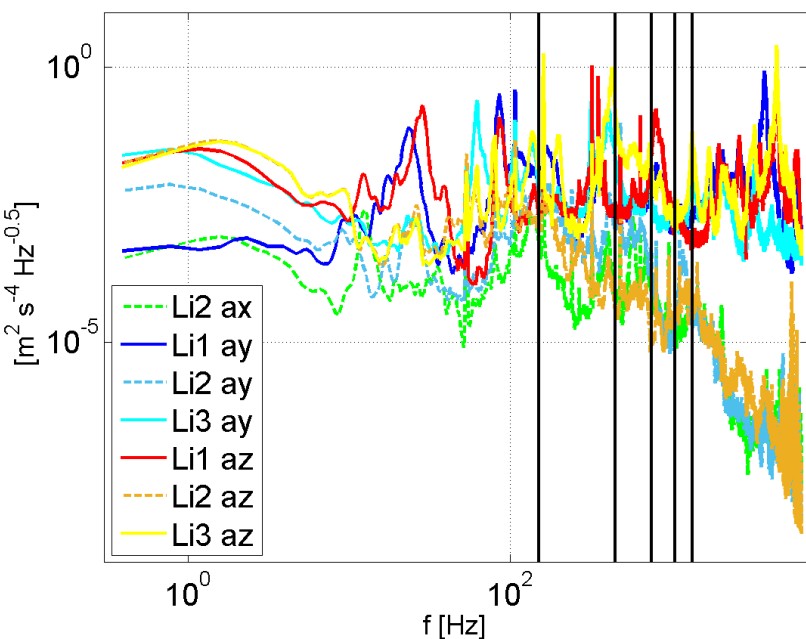

**Figure 8.** Vibration spectra measured at the three humidity sensors Li1, Li2 and Li3 during the Do128 flight on 23 October 2015, flight section D3. The colours blue and red indicate the vibrations of Li1 (y and z direction, respectively), the colours cyan and yellow indicate the vibrations of Li3 (y and z direction, respectively), and the dashed lines in green, bright blue and ocher indicate the vibrations of Li2 (x, y and z direction). The vertical black bars indicate the critical frequencies of 150 Hz and odd harmonics 450 Hz, 750 Hz, 1050 Hz, 1350 Hz according to the manufacturer.

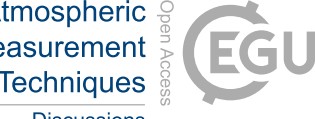



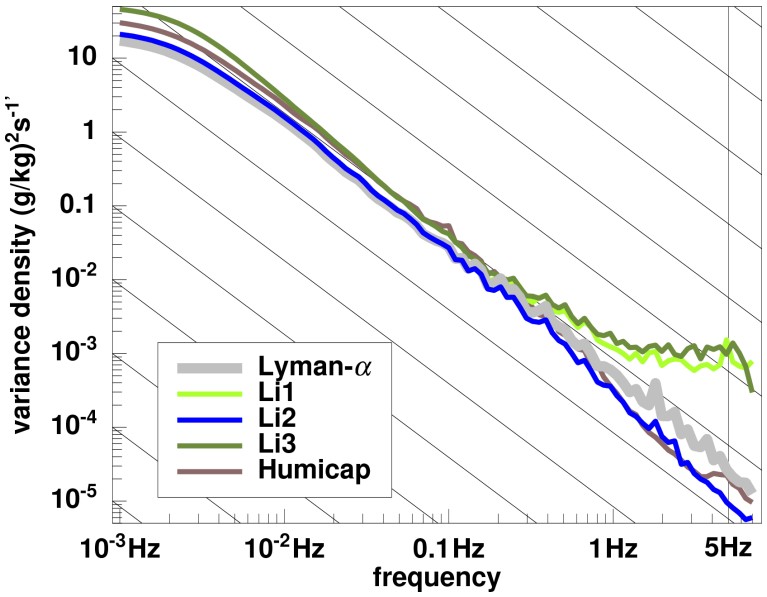

**Figure 9.** Powerspectra (variance density) of the different humidity sensors on the Do128, Li1 in bright green, Li2 in blue, Li3 in olive, Lyman-Alpha in grey, Humicap in purple. The spectra are averaged over the data of the time series shaded in grey in Fig.3.

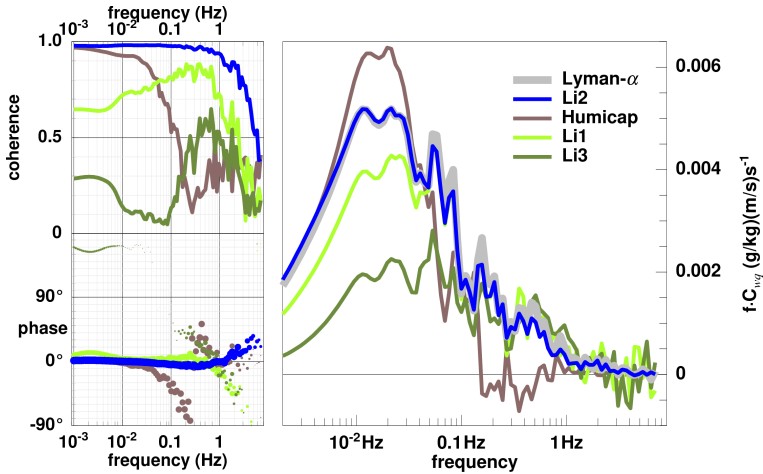

**Figure 10.** Left: Phase and coherence of the different LICOR humidity sensors on the Do128 related to the Lyman-Alpha sensor, with the same colour code as in Fig.9. In the diagram of phase shift, higher coherence is represented by larger dot size. Right: Covariance of the different humidity sensors and the vertical wind speed component on the Do128, multiplied by the frequency. The area under the curves is proportional to the humidity fluxes. The data are averaged over the data of the time series shaded in grey in Fig.3.





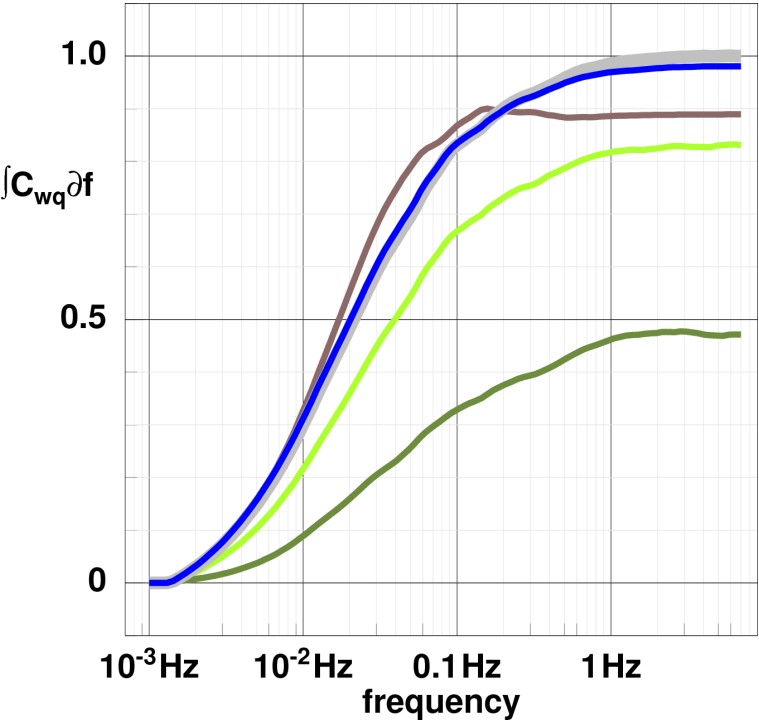

**Figure 11.** Ogive functions (integral over the latent heat fluxes) for the Do128 flight on 23 October 2015. The colours indicate the same humidity sensors as in Fig.9.





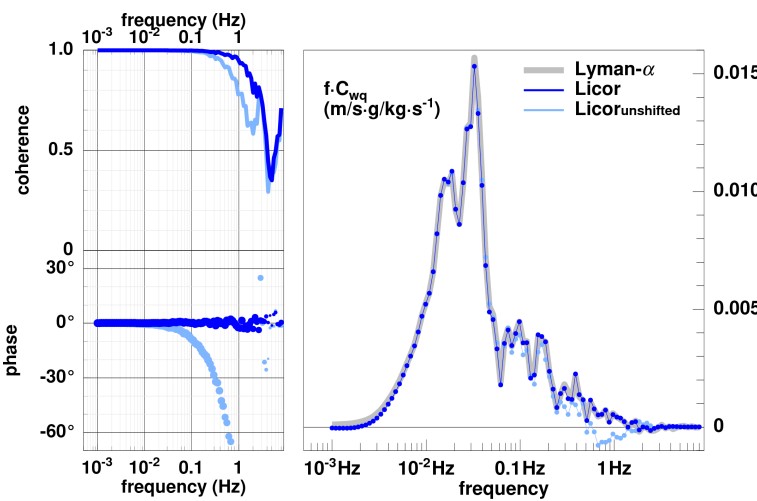

**Figure 12.** Phase shift and coherence spectra of the shifted (darker blue) and unshifted (light blue) signal of the LICOR sensor on the Helipod (left) and covariance of the different humidity sensors and the vertical wind speed component multiplied by the frequency (right) on the Helipod flight. The area under the curves is proportional to the humidity fluxes. The flight part indicated in the time series in Fig.6 in grey was used.