# Peer review of "Comparison of Lyman-Alpha and LICOR infrared hygrometers for measuring airborne turbulent fluctuations of water vapor"

_Atmospheric Measurement Techniques, 2017_

## Referee Comment (RC1) · Anonymous Referee #2 · 14 Dec 2017

Comments on the manuscript

**Comparison of the fast Lyman-Alpha and LiCor hygrometers for measuring airborne turbulent fluctuations**

Submitted by

**A. Lampert, J. Hartmann, F. Pätzold, L. Lobitz, P. Hecher, K. Kohnert, E. Larmanou, A. Serafimovich, and T. Sachs**

For publication in **Atmospheric Measurements Techniques**

The papers addresses the question, whether the infrared gas analysers LI7200/LI7500 (and followers) – which are standard instrumentation today for the measurement of turbulent water vapor and carbon dioxide concentration fluctuations and hence for the determination of the turbulent fluxes of water vapor and carbon dioxide at fixed installations (bars, masts, towers) in the surface and lower boundary layer – are as well suited for aircraft operation and might be considered as a candidate for replacing the Ly-α instruments that have been employed for airborne flux measurements over decades, but are not available on the market anymore. The authors compared the different hygrometers during two flights on two different airborne measurement platforms, the DO-128 research aircraft and the Helipod sonde. The flights have been well designed and the data analysis follows scientific standards and principles. Together with the technical-scientific relevance of the research topic this certainly justifies publication. However, I see considerable room for improving the manuscript. Some generalizations appear to be not well founded and the writing suffers from redundancy, sloppy or not very precise wording, and a rather German-language style sentence structure in several places. Detailed comments on that are given below.

**General Issues**

1.  Some statements appear too general, e.g.,
    -   P. 1, Line 20: "Measuring humidity in-situ with high accuracy is challenging". What is high accuracy? Probably this conclusion holds for any variable?
    -   P. 4, Line 11: "The Licor is the fastest and cheapest water vapor sensor commercially available" – this is too general, it is probably the cheapest fast-response sensor.
    -   P. 4, Line 15f: Is it possible to "determine the required measurement frequency for humidity fluctuations to derive reliable latent heat fluxes"? Doesn't this depend on the environment, on the turbulent state of the atmosphere, on the height of measurements, on the aircraft flight speed?
    -   P. 11, line 2f: Here the authors state that a temporal resolution of 20 Hz is sufficient for humidity flux calculations, while on p. 10, line 27, it is said that "fluctuations in the frequency range higher than 2 Hz do not contribute significantly to the overall humidity fluxes for an air speed of 70 ms$^{-1}$" – how can these two statements be brought together?

2.  Some relevant information appears not at the place where it might be expected or is missing totally. E.g., information on aircraft flight speed appears when discussing the delay times

between the sensors (instead of when describing the aircraft or the flight). Flight speed information for the Helipod flight is not given at all. The vertical wind component is needed for determining the turbulent fluxes, but no information is given on how these values were determined.

3. As a boundary layer scientist, the reader might be interested to learn something about the absolute values of the latent heat fluxes that were determined for the flights. The more since in the discussion the authors state that "especially for small fluxes, the relative errors might be significant". It is not completely clear, where this conclusion comes from. On the other side, significant relative errors for small fluxes might still mean small absolute errors, while even small relative errors for high flux values could mean significant absolute errors, which would have implications for budget studies. This aspect is not discussed in the paper, may be due to the limited representativeness of the data set.

**Specific Issues**

- The title of the paper appears somehow incomplete – the two types of hygrometers are differently named, one by the method, the other by the manufacturer, I suggest to modify it as follows: "Comparison of Lyman-Alpha and infrared hygrometers for measuring airborne turbulent fluctuations of water vapor"
- The first paragraph of the introduction consists of a series of statements which are not always in a good logical context to each other.
- P. 1, Line 14: What is "surface air moisture"? Isn't the surface layer part of the troposphere?
- P. 1, Line 15: I wonder whether Klaus et al. (2012) is really a proper reference to the difficulty of measuring and modelling global water vapor distribution.
- P. 1, Line 19: Point measurements are point measurements – what is the "horizontal extent" of a point?
- P. 1, Line 21ff: What is the relevance of the cloud chamber measurements under UTLS conditions for the present study?
- P. 2, Line 3-4: Again, two sentences with no obvious clear context: The authors want to quantify moisture transport, but speak about latent heat flux. Why not to start: The most effective way for moisture transport from the surface to the atmosphere is turbulence. Turbulent fluxes are commonly determined with the eddy-covariance method. This method requires …
- P. 2, Line 5: What are "fast fluctuations"?
- P. 2, Line 7: Isn't the high temporal resolution requested by the method, independently of whether it is used for research or not?
- P. 2, Line 9: Do the authors trust a sensor that has never been calibrated? I suggest to write "frequent re-calibration".
- P. 2, Line 9-10: Hence, there is no sensor available that meets both requirements? If this is the case you should state it.
- P. 2, Line 17: What is "sufficient humidity"?
- P. 2, Line 32: A comparative is missing here: 100 times weaker / stronger?
- P. 3, Line 1: "same order" with respect to water vapor or to the previously discussed oxygen?

- P. 3, Line 4: What is "long-term" here? May be better write "(slow) drift". Normally with long-term one would think of weeks or months, however, for Ly-α we often think of the magnitude of hours.
- P. 3, Line 10: This sounds like the need for careful calibration is a disadvantage of the KH20 which however is essentially the same for the Ly-α.
- P. 3, Line 11: What about this new sensor? If it showed promising results five years ago, is it expected to become broadly introduced? Moreover, if there is this new instrument, "THE Lyman-Alpha" (as you name it throughout the manuscript) does not exist, it could be thus wise at one place to state that "the Ly-α" in this paper is a synonym for "the Ly-α absorption hygrometer by Buck Res.".
- P. 3, Line 19-20: This sentence does not become clear here.
- P. 3, Line 22: Here the authors speak about the LICOR sensor without having introduced this. I suggest first to characterize the LICOR sensors before describing the TDLAS which is still experimental.
- P. 3, Line 26: what is "fast humidity"? Please avoid this slang in a scientific paper. Humidity is a scalar property of air, it is not fast nor slow, it is just highly variable in space in time such that you need fast-response instruments to resolve this variability.
- P. 3, Line 27: I wonder whether it is correct here (and in other places as well) to speak about a measurement chamber. LI7500 basically is an open-path sensor, even if the distance between the sensor head and base can be bridged with a "chamber". And the Lambert-Beer law underlying the physics of measurement considers the distance or length of the absorption path. Insofar, one might prefer "path" instead of "chamber".
- P. 4, Line 27f: It is not the wavelength that is absorbed but light at a wavelength of …
- P. 5, Line 11 (and also P. 6, Line 7): What makes the Humicap superior to the dew point mirror such that the latter one has not been used?
- P. 15, Line 15: What does this mean "is calibrated regularly"? How often? How?
- P. 5, Line 29f: Why has the full length of the measurement path (instead of its center length) to be considered when determining the delay time?
- P. 6, Line 18: "agricultural grassland" – do you really mean "grassland", or "farmland"?
- P. 6, Line 20ff: What was the motivation to define these six small sub-legs, knowing that the "sampling-length" requirements according to Lenschow et al. are not fulfilled here?
- P. 7, Line 1: The Ly-α data were shifted, not the instrument.
- P. 7, Line 2: Mathematically speaking the co-variance (or the correlation) is maximized.
- P. 7, Line 8, 11: On p. 5 the internal delay is given with 123 ms.
- P. 7, Line 21: The "best value" can only be one value, a range "between 0.5 and 0.9" is not a very specific information.
- P. 8, line 4: Was this the mean wind at the surface or at flight level?
- P. 8, line 14: Where is this additional time lag attributed to?
- P. 9, line 5: Why choosing a different flight section here when compared to Figure 7?
- P. 9, Line 19: Why do the authors speak of "decaying turbulence" here, the -5/3 law holds for the inertial subrange of developed turbulence.
- P. 10, line 12: … covariance of the vertical wind speed and the humidity values from the different sensors …

- P. 10, line 33: … a phase shift around 0 °C … ???
- P. 11, line 12-19: This paragraph bridles the horse from the tail. The underestimated fluxes are a consequence of the noisy, vibration-affected humidity measurement of LI1 and LI3. I suggest to organize the paragraph along this line which was followed through the paper.
- At some places, the structure of sentences is very German style, e.g., P. 1 - line 20, P. 9 – line 32f, p. 10 – line 9f, p. 11 – lines 20-21.
- In a few places there are unnecessary redundancies: P. 2 – Line 27 / Line 30, P. 4 – Line 24 / 25, caption of Figures 8, 9 (repeating the whole Figure legend).

**Figures**

- Figure 3
  - Would it be possible to indicate D1 .. D6 in the graph
  - Don't numerate the subplots without a reference, do not write "first subplot" and "main plot" etc., but "upper", "central", "lower", or label the subplots with a) … d)
  - Unit of potential temperature should be K.
  - The lower panel does not show the invalid data, instead it marks the "periods of invalid data"
  - Could the part of the flight that was used for the spectral analysis (grey shading in Figure 3, marked by a different colour in Fig. 1?
- Figure 4: This Figure is not really needed.
- Figure 6:
  - See also my remarks to Figure 3.
  - When looking at the vertical wind speed plot one gets the impression that the plot basically shows the movement of the Helipod for the ascent / descent flight periods. Shouldn't this component be removed in order to see the turbulence intensity?
- Figure 7: In fact, this plot shows the time series of accelerations in z direction which illustrate the vibrations the LICOR sensors were exposed to.
- Figure 10: Right graph should better be named as "Co-spectra of humidity from the different sensors and vertical wind speed …"
- Figure 11: I would be a bit hesitant to present a flux derived from the Humicap humidity signal in this plot without further discussion; it might be interpreted in a way that the Humicap is still much better than the vibration-sensitive LICOR instrument.

**Some minor language issues / misprints**

- P. 2, line 22: and make → making
- P. 2, line 25: with → allowing for
- P. 3, line 25: … available yet.
- P. 6, line 27: above → over
- P. 6, line 29: humidity fluxes → humidity fluctuations
- P. 8, line 1: urface → surface
- P. 9, line 24: cn → can
- P.9, line 26: sonsor → sensor

---

## Referee Comment (RC2) · Anonymous Referee #1 · 19 Dec 2017

This is a review of manuscript amt-2017-353, "Comparison of the fast Lyman-Alpha and LICOR hygrometers for measuring airborne turbulent fluctuations". The LICOR sensors in different forms are used at automated field stations for research networks covering large temporal and spatial scales, and are well characterized. The purpose of this manuscript is to evaluate LICOR humidity sensors in a new environment, on aircraft, compared with standard Lyman-alpha hygrometers. The results show that the LICOR sensors are well suited for airborne measurements of humidity fluctuations, provided that a vibrationless environment is given, and this turns out to be more important than close sensor spacing.

[Figure]

MESSAGE TO THE EDITOR: This is a detailed technical assessment of LICOR sensors that should be posted online for discussion in AMTD, and to be considered for publication only after some major revisions are made. LICOR sensors are widely used on aircraft, so validation of their performance is needed. The manuscript is an important contribution because it analyzes the environments in which the LICOR sensors perform well compared to the "gold standard" of Lyman-alpha hygrometers. It cannot be published, however, until the authors rewrite it in with better organization, better explanations and better English editing.

MAJOR CONCERN TO THE EDITOR: I still have a concern that the authors and other research groups are using LICOR sensors in an environment that the manufacturer does not recommend. Manuscript page 4, lines 9-12: The authors stated that "the manufacturer warns in the manual that the sensor should not be applied with vibrations around 150 Hz and around the harmonics". I am not an expert in this area, but would like to see more justification and proof that the LICOR is measuring accurately at 20 Hz in the aircraft environment.

The authors do not clearly explain the reason for the drift and noise in their sensors.

GENERAL and SPECIFIC COMMENTS follow in the attached supplement.

Please also note the supplement to this comment:
https://www.atmos-meas-tech-discuss.net/amt-2017-353/amt-2017-353-RC2-supplement.pdf

**Supplement:**

This is a review of manuscript amt-2017-353, "Comparison of the fast Lyman-Alpha and LICOR hygrometers for measuring airborne turbulent fluctuations" Astrid Lampert1, Jörg Hartmann2, Falk Pätzold1, Lennart Lobitz1, Peter Hecker1, Katrin Kohnert3, Eric Larmanou3,4, Andrei Serafimovich3, and Torsten Sachs3

**What follows are comments to be shared with the authors.**

**GENERAL COMMENTS:**

This is a review of manuscript amt-2017-353,

"Comparison of the fast Lyman-Alpha and LICOR hygrometers for

measuring airborne turbulent fluctuations". The LICOR sensors in different forms are used at automated field stations for research networks covering large temporal and spatial scales, and are well characterized.

The purpose of this manuscript is to evaluate LICOR humidity sensors in a new environment, on aircraft, compared with standard Lyman-alpha hygrometers. The results show that the LICOR sensors are well suited for airborne measurements of humidity fluctuations, provided that a vibrationless environment is given, and this turns out to be more important than close sensor spacing.

This is a detailed technical assessment of LICOR sensors that should be posted online for discussion in AMTD after some major revisions are made. LICOR sensors are widely used on aircraft, so validation of their performance is needed. The manuscript is an important contribution because it analyzes the environments in which the LICOR sensors perform well compared to the "gold standard" of Lyman-alpha hygrometers. I appreciate the hard work of the authors to collect the field data and carefully analyze the results. What remains for the authors to do is to rewrite the manuscript with better explanation of their reasoning and conclusions. The manuscript could also benefit from better English editing.

**SPECIFIC COMMENTS:**

1) I have a concern that the authors and other research groups are using LICOR sensors in an environment that the manufacturer does not recommend.

Manuscript page 4, lines 9-12: The authors stated that "the manufacturer warns in the manual that the sensor should not be applied with vibrations around 150 Hz and around the harmonics".

Is it possible for the authors to contact the LICOR manufacturer to get approval - or feedback from the Technical Support department - for flying a LICOR on aircraft?

2) The reported experiment involved one vibration-isolated closed-path hygrometer and two non-isolated open-path hygrometers, so how do you know whether the drift and noise are due to vibration or the open-path? Are there other possible reasons for the drift (such as drift in the electronics response or internal processing?)?

3) Section 1.3, page 3, lines 14-25, describe laser hygrometers but has some gaps as listed below:

3.1) Page 3, lines 24-25 claim that "it is not possible to obtain real-time humidity data." Although the Buchholz et al. hygrometer (Buchholz et al. 2014) does not provide real-time humidity data, other laser hygrometers do this routinely (see papers such as S. B. Smith et al., JGR, 2017, R. L. Herman et al., ACP, 2017, or M. Zondlo et al.).

3.2) page 3, line 25 claims that "The spectroscopic sensors are experimental systems and not commercially available" but the Picarro and Los Gatos systems mentioned earlier are commercially available laser hygrometers that have sufficient accuracy for the science. They can also provide real-time humidity data.

3.3) page 4, lines 10-11 claims that "...the LICOR sensor is currently the fastest and cheapest water vapor sensor commercially available" but laser hygrometers are faster than the LICOR.

4) I find the discussion of the time resolution of the LICOR to be disorganized and confusing (Sections 2.1 and 2.2 and 3.4). I recommend that the authors should reorganize the discussion of the time response, time delay and time synchronization to one section because these are related.

4.1) What is the time resolution of the LICOR instrument? Page 5, line 2, indicates that the data is "internally processing and finally provided at a maximum frequency of 20 Hz." Are the detector and electronics signal chain sufficiently fast to resolve changes in water vapor at 20 Hz?

4.2) Page 5, line 9: what is the response time of the Rosemount EL102 sensor? It is only characterized here as a "fast response time." How fast?

4.3) Page 5, lines 25-30: How can you carry out successful fast measurements with the closed-path LICOR if there is a 250-millisecond calculated delay? Have you tested the delay? What is the residence time in the sample cell?

4.4) Page 10, line 23 and Page 11, line 25: the authors state the "Generally a temporal resolution of 20 Hz is sufficient for humidity flux calculations." It is not clearly explained how the authors came to this conclusion. Is there a reference that can be cited as evidence for this? Furthermore, it is unclear from this manuscript whether the LICOR has an actual temporal resolution of 20 Hz (when the sampling delays and internal processing are included).

5) Flight figures 3 and 6 are hard to read: please consider larger font text.

---

## Author Comment (AC1) · 12 Feb 2018

**Answers to Referee 1: Comparison of the fast Lyman-Alpha and LICOR hygrometers for measuring airborne turbulent fluctuations**

Astrid Lampert[1], Jörg Hartmann[2], Falk Pätzold[1], Lennart Lobitz[1], Peter Hecker[1], Katrin Kohnert[3], Eric Larmanou[3,4], Andrei Serafimovich[3], and Torsten Sachs[1,3]

[1]Institute of Flight Guidance, TU Braunschweig, Hermann-Blenk-Str. 27, 38108 Braunschweig, Germany
[2]Alfred Wegener Institute for Polar and Marine Research, Bussestr. 24, 27570 Bremerhaven, Germany
[3]GFZ German Research Centre for Geosciences, Telegrafenberg, 14473 Potsdam, Germany
[4]Swedish University of Agricultural Sciences, Umeå, Sweden

*Correspondence to:* Astrid Lampert (Astrid.Lampert@tu-braunschweig.de)

The authors would like to thank the anonymous referee for the thorough review with very detailed comments, which helped to improve the manuscript significantly. In the following, each comment of the referee (in italic) is answered separately. The answers are provided in normal style, and the changed text of the manuscript is given in quotation marks.

*The papers addresses the question, whether the infrared gas analysers LI7200/LI7500 (and followers) – which are standard*
5   *instrumentation today for the measurement of turbulent water vapor and carbon dioxide concentration fluctuations and hence for the determination of the turbulent fluxes of water vapor and carbon dioxide at fixed installations (bars, masts, towers) in the surface and lower boundary layer – are as well suited for aircraft operation and might be considered as a candidate for replacing the Ly-Alpha instruments that have been employed for airborne flux measurements over decades, but are not available on the market anymore. The authors compared the different hygrometers during two flights on two different airborne*
10   *measurement platforms, the DO-128 research aircraft and the Helipod sonde. The flights have been well designed and the data analysis follows scientific standards and principles. Together with the technical-scientific relevance of the research topic this certainly justifies publication. However, I see considerable room for improving the manuscript. Some generalizations appear to be not well founded and the writing suffers from redundancy, sloppy or not very precise wording, and a rather German-language style sentence structure in several places. Detailed comments on that are given below.*

15   The authors appreciate the positive comments about the content of the manuscript. We would like to thank the referee for the great effort, and agree that there is room for improving the language and style. This is done by taking into account the very detailed cocmments of the referees, and generally by critically proof-reading the manuscript again. In the following, we will answer each comment separately.

*General Issues 1. Some statements appear too general, e.g., - P. 1, Line 20: "Measuring humidity in-situ with high accuracy*
20   *is challenging". What is high accuracy? Probably this conclusion holds for any variable?*

We agree with the referee that this can be said about any variable. However, the error bar for water vapour measurements is much larger than for e.g. temperature. We changed the text to:

"For *in situ* measurements of humidity, the error bars are typically larger than for other atmospheric parameters like temperature and wind."

*- P. 4, Line 11: "The Licor is the fastest and cheapest water vapor sensor commercially available" – this is too general, it is probably the cheapest fast-response sensor.*

We changed the text to:// "As the LICOR sensor is currently the cheapest fast-response water vapour sensor commercially available"

*- P. 4, Line 15f: Is it possible to "determine the required measurement frequency for humidity fluctuations to derive reliable latent heat fluxes"? Doesn't this depend on the environment, on the turbulent state of the atmosphere, on the height of measurements, on the aircraft flight speed?*

We changed the text to:

"to determine the required measurement frequency for humidity fluctuations to derive reliable latent heat fluxes for the typical flight altitude of few 100 m and airspeed in the range of 35 to 70 m s$^{-1}$"

*- P. 11, line 2f: Here the authors state that a temporal resolution of 20 Hz is sufficient for humidity flux calculations, while on p. 10, line 27, it is said that "fluctuations in the frequency range higher than 2 Hz do not contribute significantly to the overall humidity fluxes for an air speed of 70 m s-1" – how can these two statements be brought together?*

To make it clearer, we changed the concluding sentence to:

"Generally the temporal resolution of the LICOR sensors of 20 Hz is sufficient for humidity flux calculations, as the contribution of frequencies above approximately 2 Hz is negligible, so a 10 times oversampling for a sufficient amplitude retrieval is provided."

*2. Some relevant information appears not at the place where it might be expected or is missing totally. E.g., information on aircraft flight speed appears when discussing the delay times between the sensors (instead of when describing the aircraft or the flight). Flight speed information for the Helipod flight is not given at all. The vertical wind component is needed for determining the turbulent fluxes, but no information is given on how these values were determined.*

We included the information on the flight speed at the beginning of the section describing the flights. For the Do128: "The measurement flight with the research aircraft Do128 "D-IBUF" was conducted on 23 October 2015. The aircraft operates at a true airspeed of 70 m s$^{-1}$. The flight was performed above different terrain of the North German Plain," For the Helipod: "During the measurement flight, the Helipod was attached to a Russian Mi8 helicopter by a 30 m rope. The flight was performed at a true airspeed of 40 m s$^{-1}$ from the Research Station Samoylov Island in the Lena Delta, Siberia."

Determining the 3D wind speed from five-hole probe, GNSS and inertial data is quite complex, but a standard method, which is not the focus of this article. Therefore we included two additional references in the text:

"a five-hole probe and corresponding pressure transducers of Setra (static, dynamic and differential pressure), inertial navigation and global navigation satellite system (GNSS) for deriving the 3D wind vector (see description of method e.g. van den Kroonenberg et al., 2008; Bärfuss et al., 2018)"

*3. As a boundary layer scientist, the reader might be interested to learn something about the absolute values of the latent heat fluxes that were determined for the flights. The more since in the discussion the authors state that "especially for small fluxes, the relative errors might be significant". It is not completely clear, where this conclusion comes from. On the other side, significant relative errors for small fluxes might still mean small absolute errors, while even small relative errors for high flux*

*values could mean significant absolute errors, which would have implications for budget studies. This aspect is not discussed in the paper, may be due to the limited representativeness of the data set.*

In the article, we prefer to focus on the instrumental comparison, which is the basis for retrieving latent heat fluxes. The application of the sensors for boundary layer studies, and in particular a detailed analysis of the Helipod flights in Siberia, require a complete description of the atmospheric conditions, and are beyond the scope of the article. Articles about the data set are currently under preparation.

*Specific Issues*

*- The title of the paper appears somehow incomplete – the two types of hygrometers are differently named, one by the method, the other by the manufacturer, I suggest to modify it as follows: "Comparison of Lyman-Alpha and infrared hygrometers for measuring airborne turbulent fluctuations of water vapor"*

We changed the title to "Comparison of Lyman-Alpha and LICOR infrared hygrometers for measuring airborne turbulent fluctuations of water vapor". The title suggested by the referee would include the discussion of other types of infrared hygrometers as well, which we do not provide.

*- The first paragraph of the introduction consists of a series of statements which are not always in a good logical context to each other.*

We re-arranged the text and tried to establish a logical order of the thoughts:

"Water vapour and clouds in the atmosphere have a large impact on the energy balance (**?**), the hydrologic cycle (e.g. **?**) and on local and global climate (**??**). Therefore, accurate knowledge about atmospheric water vapour is of high relevance for understanding climate and climate change. A general increase in atmospheric moisture measured at the surface and humidity within the troposphere has been reported (**?**). Satellite retrievals of the vertical water vapour distribution provide limited spatial resolution, e.g. 300 m in the vertical and 30 km in horizontal direction (**?**). For the quantification of atmospheric processes on local to regional scales, airborne measurements are required to fill the gap between large-scale, low resolution information from satellites and point measurements with higher vertical and temporal resolution, but limited in horizontal extent."

*- P. 1, Line 14: What is "surface air moisture"? Isn't the surface layer part of the troposphere?*

To clarify, we changed the sentence to:

"A general increase in atmospheric moisture measured at the surface and humidity within the troposphere has been reported"

*- P. 1, Line 15: I wonder whether Klaus et al. (2012) is really a proper reference to the difficulty of measuring and modelling global water vapor distribution.*

We removed the sentence with the misleading reference (However, the global distribution of moisture is difficult to measure and model accurately due to its large spatial and temporal variability (e.g. Klaus et al., 2012).). Probably it is not necessary to talk about global moisture distribution, when the scope of the article are measurements with high resolution.

*- P. 1, Line 19: Point measurements are point measurements – what is the "horizontal extent" of a point?*

We changed the text to:

"measurements at fixed locations with higher vertical and temporal resolution, but representative only for a small area"

*- P. 1, Line 21ff: What is the relevance of the cloud chamber measurements under UTLS conditions for the present study?*

We would like to emphasize that measuring atmospheric water vapour precisely is difficult, and even with the best systems under well controlled conditions in the laboratory, there are large discrepancies between different measurement systems. We changed the text to:

"The uncertainties of atmospheric water vapour measurements are high, as even for well controlled conditions in a cloud chamber, intercomparison measurements of different hygrometers probing the same air simultaneously revealed discrepancies between different measurement systems of around $10\,\%$"

*- P. 2, Line 3-4: Again, two sentences with no obvious clear context: The authors want to quantify moisture transport, but speak about latent heat flux. Why not to start: The most effective way for moisture transport from the surface to the atmosphere is turbulence. Turbulent fluxes are commonly determined with the eddy-covariance method. This method requires . . .*

We changed the text to:

"The most effective way for moisture transport from the surface to the atmosphere is turbulence. Turbulent fluxes are commonly determined with the eddy-covariance method. This technique requires accurate measurements of the fluctuations of the vertical component of wind speed and humidity."

*- P. 2, Line 5: What are "fast fluctuations"?*

True - we removed "fast".

*- P. 2, Line 7: Isn't the high temporal resolution requested by the method, independently of whether it is used for research or not?*

We changed the text to:

"Airborne sensors have to fullfil specific requirements. On the one hand, a high temporal resolution is needed in order to obtain a high spatial resolution for the moving platforms. On the other hand, long-term stability and high accuracy, if possible without the need of frequent re-calibration, are essential."

*- P. 2, Line 9: Do the authors trust a sensor that has never been calibrated? I suggest to write "frequent re-calibration".*

We agree, and changed as suggested.

*- P. 2, Line 9-10: Hence, there is no sensor available that meets both requirements? If this is the case you should state it.*

We changed the text to:

"In practice, as no sensor is available that meets both requirements, this leads to the combination of complementary sensors for both high resolution and long-term accuracy."

*- P. 2, Line 17: What is "sufficient humidity"?*

As this is not quantified in the reference, we removed it from the sentence, and changed the text to:

"For temperatures exceeding $0\,°C$, and with the help of extensive postprocessing or modelling, the relatively slow polymer-based absorption hygrometers are sometimes used for retrieving humidity fluctuations"

*- P. 2, Line 32: A comparative is missing here: 100 times weaker / stronger?*

Thank you for the remark! When verifying this point, we saw that there was even a mistake in the order of magnitude of the effect. The sentence has been changed to:

"The absorption by oxygen molecules is about 1000 times weaker than by ozone molecules, and can be corrected by taking into account pressure and temperature, as the fractional density is constant."

*- P. 3, Line 1: "same order" with respect to water vapor or to the previously discussed oxygen?*

5     We changed the text to:

"same order of magnitude with respect to water vapour"

*- P. 3, Line 4: What is "long-term" here? May be better write "(slow) drift". Normally with long-term one would think of weeks or months, however, for Ly-Alpha we often think of the magnitude of hours.*

    We changed as suggested.

10    *- P. 3, Line 10: This sounds like the need for careful calibration is a disadvantage of the KH20 which however is essentially the same for the Ly-Alpha.*

    We changed the text to:

"A similar system is the Krypton hygrometer KH20 of Campbell Scientific, USA, which has a cross sensitivity to oxygen as well and therefore, like the Lyman-Alpha, has to be calibrated carefully"

15    *- P. 3, Line 11: What about this new sensor? If it showed promising results five years ago, is it expected to become broadly introduced? Moreover, if there is this new instrument, "THE Lyman-Alpha" (as you name it throughout the manuscript) does not exist, it could be thus wise at one place to state that "the Ly-Alpha" in this paper is a synonym for "the Ly-Alpha absorption hygrometer by Buck Res.".*

    Very good idea! As suggested, we included in Sect. 1.2 the following sentence:

20 "The term "Lyman-Alpha" in this paper is used as a synonym for the Lyman-Alpha absorption hygrometer by Buck Research". Concerning the KH20: We did not find more literature about airborne applications, it seems to be mostly used for ground-based measurements. According to Foken and Falke (2010), the instrument is very sensitive to path length, and calibration is difficult even for ground-based applications. We added in the text:

"It is, however, not broadly present in airborne applications."

25    *- P. 3, Line 19-20: This sentence does not become clear here.*

    We re-phrased the sentence:

"For retrieving methane and carbon dioxide with these instruments, the water vapour measurements are necessary to reference the number concentration of methane molecules to the dry mole fraction."

*- P. 3, Line 22: Here the authors speak about the LICOR sensor without having introduced this. I suggest first to characterize*

30 *the LICOR sensors before describing the TDLAS which is still experimental.*

    We changed the order of introducing the sensors. Now we first present the LICOR sensors and measurement principle, then the TDLAS.

*- P. 3, Line 26: what is "fast humidity"? Please avoid this slang in a scientific paper. Humidity is a scalar property of air, it is not fast nor slow, it is just highly variable in space in time such that you need fast-response instruments to resolve this variability.*

We changed the text to:

"The fast-response LICOR instruments LI-7500, LI-7500A and LI-7200 for measuring humidity "

5    *- P. 3, Line 27: I wonder whether it is correct here (and in other places as well) to speak about a measurement chamber. LI7500 basically is an open-path sensor, even if the distance between the sensor head and base can be bridged with a "chamber". And the Lambert-Beer law underlying the physics of measurement considers the distance or length of the absorption path. Insofar, one might prefer "path" instead of "chamber".*

We agree that "path" is the better suited expression and changed that in the manuscript.

10    *- P. 4, Line 27f: It is not the wavelength that is absorbed but light at a wavelength of . . .*

We changed this as suggested.

*- P. 5, Line 11 (and also P. 6, Line 7): What makes the Humicap superior to the dew point mirror such that the latter one has not been used?*

During the Do-128 flight, the dew point mirror was not operating properly. Therefore, we prefer to show only the results of

15    the same sensor types for the Helipod as well, which are the Humicap, Lyman-Alpha and LICOR sensors.

*- P. 15, Line 15: What does this mean "is calibrated regularly"? How often? How?*

We changed the text to:

"The Vaisala Humicap is calibrated before and after each measurement campaign by applying saturated salt solutions and their different known equilibrium relativ humidity"

20    *- P. 5, Line 29f: Why has the full length of the measurement path (instead of its center length) to be considered when determining the delay time?*

Thank you for the hint, that is correct. We use indeed the center length for the calculation resulting in a time delay of 0.12 s, not the full length of the measurement path. In the text, this is not clear. Therefore we changed the text to:

"the time for exchanging the air of the measurement cell with an inner diameter of 25 mm and a half length of 125 mm amounts

25    to 0.12 s."

*- P. 6, Line 18: "agricultural grassland" – do you really mean "grassland", or "farmland"?*

We changed the text to "agricultural farmland".

*- P. 6, Line 20ff: What was the motivation to define these six small sub-legs, knowing that the "sampling-length" requirements according to Lenschow et al. are not fulfilled here?*

30    We chose these small sub-legs with different but homogeneous surface conditions and different but constant flight altitudes to compare if there are systematic differences in the important parameters like the vibration level. We added in the text:

"These small sub-legs were chosen with different but homogeneous surface conditions and different but constant flight altitudes to compare if there are systematic differences in the parameters like the vibration level."

*- P. 7, Line 1: The Ly-Alpha data were shifted, not the instrument.*

35    We changed the text to:

" First the Lyman-Alpha data were shifted in time"

*- P. 7, Line 2: Mathematically speaking the co-variance (or the correlation) is maximized.*

We changed the text to:

"The synchronisation was then done by maximizing the covariance of the mixing ratio fluctuations of the Lyman-Alpha and
each of the LICOR sensors"

*- P. 7, Line 8, 11: On p. 5 the internal delay is given with 123 ms.*

Thank you for pointing out this mistake. We verified with the manuals that the internal delay time of 130 ms is correct, as on
p. 7, which is used for the calculations. We corrected the value of the internal delay given on p. 5.

*- P. 7, Line 21: The "best value" can only be one value, a range "between 0.5 and 0.9" is not a very specific information.*

We changed the text to:

"The best correlation between Li3 and Lyman-Alpha amounted to 0.6, thus was considerably lower (Fig. 10)"

*- P. 8, line 4: Was this the mean wind at the surface or at flight level?*

To clarify, we changed the text to:

"The flight on 14 August 2014 was done in conditions nearly free of clouds at the beginning with a near-surface air temperature
around 17 °C and southerly wind with a speed of $5 \, \mathrm{m \, s^{-1}}$ near ground. The mean wind speed at the altitude of the Helipod
transects was $8 \, \mathrm{m \, s^{-1}}$, and the mean wind direction at that altitude was 180 °."

*- P. 8, line 14: Where is this additional time lag attributed to?*

We added in the text:

"The additional time lag may be attributed to the semi-open housing geometry."

*- P. 9, line 5: Why choosing a different flight section here when compared to Figure 7?*

We changed that. Now Fig. 7 and Fig. 8 both show flight section D3.

*- P. 9, Line 19: Why do the authors speak of "decaying turbulence" here, the -5/3 law holds for the inertial subrange of
developed turbulence.*

To avoid confusion, we changed the text to:

" The sloped lines represent the -5/3 drop-off expected in the inertial subrange."

*- P. 10, line 12: . . . covariance of the vertical wind speed and the humidity values from the different sensors . . .*

We changed as suggested.

*- P. 10, line 33: . . . a phase shift around 0 °C . . . ???*

We changed the text to:// "and the phase shift around 0 °"

*- P. 11, line 12-19: This paragraph bridles the horse from the tail. The underestimated fluxes are a consequence of the
noisy, vibration-affected humidity measurement of LI1 and LI3. I suggest to organize the paragraph along this line which was
followed through the paper.*

We changed the text to:

"For the Do128 application, three different LICOR sensors were subject to different vibration levels. For the Li1 and Li3
sensors, installed without particular isolation agains vibrations, the correlation with the Lyman-Alpha signal was significantly
lower than for the Li2 sensor, which was installed isolated against vibrations. The different covariance spectra of the vibration-
affected humidity measurements of the Li1 and Li3 sensors resulted in larger deviations of the latent heat fluxes compared to

the latent heat fluxes based on the Lyman-Alpha sensor. The vibration-isolated Li2 sensor showing high correlation with the Lyman-Alpha sensor resulted in comparable latent heat fluxes. However, the spectral behaviour of the vibrations had no direct,

5    linear impact on the humidity spectra of the Li1 and Li3 sensors, but the relationship is more complex. This is currently subject to more detailed investigations."

*- At some places, the structure of sentences is very German style, e.g., P. 1 - line 20, P. 9 – line 32f, p. 10 – line 9f, p. 11 – lines 20-21.*

We changed the sentences to:

10   "For *in situ* measurements of humidity, the error bars are typically larger than for other atmospheric parameters like temperature and wind."

"Based on the grey shaded data set of Fig. 3, Fig. 10 shows the coherence and the phase of the different LICOR sensors with the Lyman-Alpha. Overall, the Li2 provides the best coherence with the Lyman-Alpha. The coherence is virtually equal to one over a large frequency range of three decades. It only drops off for frequencies beyond 1 Hz due to the spatial separation of the

15   two sensors. No phase difference is observed over the same frequency range."

"The response behaviour of the Vaisala Humicap is more complex. At low frequencies (<0.01 Hz) it agrees reasonably well with the Lyman-Alpha. Then the coherence decreases with increasing frequency. The phase shift disappears around 0.4 Hz, but the level of coherence remains lower."

"For the Helipod application with lower vibrations, the humidity fluxes derived from the Lyman-Alpha and LICOR sensors

20   agreed very well after careful sensor calibration to absolute values, and correction of the time lag."

*- In a few places there are unnecessary redundancies: P. 2 – Line 27 / Line 30, P. 4 – Line 24 / 25, caption of Figures 8, 9 (repeating the whole Figure legend).*

We removed the redundancies for the first two parts. The sentences were changed to://

"Sensors based on atomic absorption provide the advantage of a very fast response time allowing for measurement frequen-

25   cies exceeding 100 Hz, a sharp absorption line compared to the absorption bands of molecules, and a high degree of absorption. This requires only measurement cells of few mm (**?**) compared to several cm for molecular absorption."

"The working principle of the LICOR sensor series for water vapour and carbon dioxide ($CO_2$) is the absorption of near infrared radiation by these molecules."

30

However, we are not sure which figure captions the referee refers to, as Fig. 8 and 9 represent totally different parameters. When applicable, figure captions already refer to the colour schemes of previous figures. For Fig. 10 and 12, as well as Fig. 3 and 6, we prefer to repeat the figure caption, as there are the data of different sensors, displayed in different colours.

*Figures*

*- Figure 3*

*- Would it be possible to indicate D1 .. D6 in the graph*

As we perform the spectral analysis with the part of the data shaded in grey, we prefer not to include the flight legs D1 to D5. They were mainly used for comparison studies.

5    *- Don't numerate the subplots without a reference, do not write "first subplot" and "main plot" etc., but "upper", "central", "lower", or label the subplots with a) ... d)*

We labelled the subplots as suggested.

*- Unit of potential temperature should be K.*

We changed to the unit K.

10    *- The lower panel does not show the invalid data, instead it marks the "periods of invalid data"*

We changed that.

*- Could the part of the flight that was used for the spectral analysis (grey shading in Figure 3, marked by a different colour in Fig. 1?*

The referee probably refers to the flight path in Fig. 2. We implemented this in the map.

15    *- Figure 4:*

*This Figure is not really needed.*

We removed Fig. 4 and removed the reference in the text.

*- Figure 6:*

*- See also my remarks to Figure 3.*

20    We adapted the same points as mentioned above for Fig. 6 as well.

*- When looking at the vertical wind speed plot one gets the impression that the plot basically shows the movement of the Helipod for the ascent / descent flight periods. Shouldn't this component be removed in order to see the turbulence intensity?*

The ascent and descent parts of the flight were done in spirals, with a reduced true air speed of $30\,\mathrm{m\,s^{-1}}$ instead of the $40\,\mathrm{m\,s^{-1}}$ required to escape the downwash effects of the helicopter, and a banking angle of $15\,^{\circ}$. Therefore we use only the

25    straight and level flight legs for the instrumental comparison.

*- Figure 7: In fact, this plot shows the time series of accelerations in z direction which illustrate the vibrations the LICOR sensors were exposed to.*

True, we changed to "accelerations". We further added in the section about the Do-128: "The axis of the optical path of all sensors were oriented along the aircraft longitudinal axis."

30    *- Figure 10: Right graph should better be named as "Co-spectra of humidity from the different sensors and vertical wind speed ... "*

We changed the expression for Fig. 10 and 12 as suggested.

*- Figure 11: I would be a bit hesitant to present a flux derived from the Humicap humidity signal in this plot without further discussion; it might be interpreted in a way that the Humicap is still much better than the vibration-sensitive LICOR instrument.*

35    Indeed the figure shows that in this case, the Humicap is better suited for determining latent heat fluxes than the vibration affected LICOR sensors. We added in the discussion:

"The latent heat flux determined with the Humicap amounts to 95% of the reference value determined with the Lyman-Alpha.

This means that for moderate conditions (10-20°C, humidity values typical for midlatitudes), the Humicap can be used for determining airborne latent heat fluxes with an acceptable error bar. However, the response function of the Humicap is asymmetric, with a different response time for decreasing and increasing humidity, and the response time becomes significantly slower for cold conditions like in the Arctic, where the sensor is not suitable for deriving latent heat fluxes."

*Some minor language issues / misprints*

*- P. 2, line 22: and make ... making*

*- P. 2, line 25: with ... allowing for*

*- P. 3, line 25: . . . available yet.*

*- P. 6, line 27: above ... over*

*- P. 6, line 29: humidity fluxes ... humidity fluctuations*

*- P. 8, line 1: urface ... surface*

*- P. 9, line 24: cn ... can*

*- P.9, line 26: sonsor ... sensor* We thank the referee for careful reading and corrected the mentioned points.

**References**

Bärfuss, K., Pätzold, F., Altstädter, B., Kathe, E., Nowak, S., Bretschneider, L., Bestmann, U., and Lampert, A.: New Setup of the UAS ALADINA for Measuring Boundary Layer Properties, Atmospheric Particles and Solar Radiation, Atmosphere, 9, 28; doi:10.3390/atmos9010028, 21 pp., 2018.

Buck, A. L.: Development of an improved Lyman-alpha hygrometer, Atmos. Technol., 2, 213-240, 1973.

5    Foken, T., and Falke, F.: Documentation and Instruction Manual for the Krypton Hygrometer Calibration Instrument, Technical Report, https://epub.uni-bayreuth.de/390/1/ARBERG042.pdf, 21 pp., 2010.

Klaus, D., Dorn, W., Dethloff, K., Rinke, A., and Mielke, M.: Evaluation of Two Cloud Parameterizations and Their Possible Adaptation to Arctic Climate Conditions, Atmosphere, 3, 419-450, 2012.

van den Kroonenberg A.C., Martin T., Buschmann M., Bange J., and Vörsmann, P.: Measuring the Wind Vector Using the Autonomous Mini

10    Aerial Vehicle $M^2AV$, J. Atmos. Ocean. Technolog., 25, 1969-1982, 2008.

---

## Author Comment (AC2) · 12 Feb 2018

**Answers to Referee 2: Comparison of the fast Lyman-Alpha and LICOR hygrometers for measuring airborne turbulent fluctuations**

Astrid Lampert[1], Jörg Hartmann[2], Falk Pätzold[1], Lennart Lobitz[1], Peter Hecker[1], Katrin Kohnert[3], Eric Larmanou[3,4], Andrei Serafimovich[3], and Torsten Sachs[1,3]

[1]Institute of Flight Guidance, TU Braunschweig, Hermann-Blenk-Str. 27, 38108 Braunschweig, Germany
[2]Alfred Wegener Institute for Polar and Marine Research, Bussestr. 24, 27570 Bremerhaven, Germany
[3]GFZ German Research Centre for Geosciences, Telegrafenberg, 14473 Potsdam, Germany
[4]Swedish University of Agricultural Sciences, Umeå, Sweden

*Correspondence to:* Astrid Lampert (Astrid.Lampert@tu-braunschweig.de)

The authors would like to thank the anonymous referee for the suggestions and corrections. In the following, each comment of the referee (in italic) is answered separately. The answers are provided in normal style, and the changed text of the manuscript is given in quotation marks.

*GENERAL COMMENTS: This is a review of manuscript amt-2017-353, "Comparison of the fast Lyman-Alpha and LICOR*

5 *hygrometers for measuring airborne turbulent fluctuations". The LICOR sensors in different forms are used at automated field stations for research networks covering large temporal and spatial scales, and are well characterized. The purpose of this manuscript is to evaluate LICOR humidity sensors in a new environment, on aircraft, compared with standard Lyman-alpha hygrometers. The results show that the LICOR sensors are well suited for airborne measurements of humidity fluctuations, provided that a vibrationless environment is given, and this turns out to be more important than close sensor spacing. This*

10 *is a detailed technical assessment of LICOR sensors that should be posted online for discussion in AMTD after some major revisions are made. LICOR sensors are widely used on aircraft, so validation of their performance is needed. The manuscript is an important contribution because it analyzes the environments in which the LICOR sensors perform well compared to the "gold standard" of Lyman-alpha hygrometers. appreciate the hard work of the authors to collect the field data and carefully analyze the results.*

15 We would like to thank the referee for acknowledging the importance of the work presented in the manuscript.

*What remains for the authors to do is to rewrite the manuscript with better explanation of their reasoning and conclusions.*

We included more explanations in the manuscript, which take into account the comments of the referees.

*The manuscript could also benefit from better English editing.*

The English style has been improved by taking into account the very detailed comments of Referee 1. Several sentences

20 have been re-written, and the grammatical and spelling mistakes have been corrected.

*SPECIFIC COMMENTS: 1 I have a concern that the authors and other research groups are using LICOR sensors in an environment that the manufacturer does not recommend.*

We understand that the manufacturer does not sell the system for airborne applications, as they are aware that vibrations may hamper the data. For that reason, we consider it even more important to figure out the limitations of the sensor in terms of vibrations.

*Manuscript page 4, lines 9-12: The authors stated that "the manufacturer warns in the manual that the sensor should not be applied with vibrations around 150 Hz and around the harmonics". Is it possible for the authors to contact the LICOR manufacturer to get approval - or feedback from the Technical Support department - for flying a LICOR on aircraft?*

Yes, we were in contact with the manufacturer, and presented the results. Some stafff members showed large interest in the investigations, however, we did not get an official statement from the company.

*2 The reported experiment involved one vibration-isolated closed-path hygrometer and two non-isolated open-path hygrometers, so how do you know whether the drift and noise are due to vibration or the open-path? Are there other possible reasons for the drift (such as drift in the electronics response or internal processing?)?* We operated the identical open path sensor on a platform affected by vibrations (Do128, not isolated), and on a platform with much lower level of vibrations (Helipod, carried by helicopter, without own propulsion). For the latter application, the open-path sensor performed well without any drift. This comparison allows the conclusion that the vibrations are responsible for degrading the measurements. To clarify, we added the following sentence in the conclusion section:

"Altogether, both open-path and closed-path LICOR sensors are suitable high-resolution hygrometers for airborne applications, if the vibrations are low."

*3 Section 1.3, page 3, lines 14-25, describe laser hygrometers but has some gaps as listed below: 3.1) Page 3, lines 24-25 claim that "it is not possible to obtain real-time humidity data." Although the Buchholz et al. hygrometer (Buchholz et al. 2014) does not provide real-time humidity data, other laser hygrometers do this routinely (see papers such as S. B. Smith et al., JGR, 2017, R. L. Herman et al., ACP, 2017, or M. Zondlo et al.).*

We added in the text:

"For large research aircraft, some specifically designed hygrometers are implemented: On the National Science Foundation Gulfstream-V aircraft, a cavity diode laser hygrometer with two absorption lines in the near-infrared is deployed with a temporal resolution of 25 Hz (Zondlo et al., 2010). On the NASA ER-2 aircraft, a specifically designed near-infrared tunable diode laser spectrometer is deployed for measuring atmospheric water vapour concentration (May, 1998), with a sampling rate of 1 Hz and 10% accuracy(Herman et al., 2017). Compared to the LICOR sensor, this tunable diode laser hygrometer can be operated much faster (up to several kHz) and with a known accuracy, providing the most precise humidity values available to date (Buchholz et al., 2013, 2014, 2016). However, this hygrometer requires extensive post processing, and at least so far it is not possible to obtain real-time humidity data. The spectroscopic sensors are experimental systems and not commercially available yet."

*3.2 page 3, line 25 claims that "The spectroscopic sensors are experimental systems and not commercially available" but the Picarro and Los Gatos systems mentioned earlier are commercially available laser hygrometers that have sufficient accuracy for the science. They can also provide realtime humidity data.* Yes, we agree with the referee. However, the price for a Picarro and Los Gatos sensor is much higher, and so is the weight, which makes them not so easy to use in aircraft. We changed the

text to:

"As the LICOR sensor is currently the cheapest fast-response water vapour sensor commercially available, and small enough to be easily integrated into aircraft, its airborne applications will very likely increase."

*3.3 page 4, lines 10-11 claims that "... the LICOR sensor is currently the fastest and cheapest water vapor sensor commercially available" but laser hygrometers are faster than the LICOR.* We changed the text to:

"the LICOR sensor is currently the cheapest fast-response water vapour sensor commercially available"

*4 I find the discussion of the time resolution of the LICOR to be disorganized and confusing (Sections 2.1 and 2.2 and 3.4). I recommend that the authors should reorganize the discussion of the time response, time delay and time synchronization to one section because these are related.*

We are not very happy with this suggestion. Section 2.1 generally introduces the LICOR sensors with their properties. However, the time synchronisation stronly depends on the setup in the corresponding measurement platform with the tube lengths, and distances between hygrometers and 5-hole probe. Therefore, we would prefer to discuss the synchronisation separately for each airborne carrier platform.

*4.1 What is the time resolution of the LICOR instrument? Page 5, line 2, indicates that the data is "internally processing and finally provided at a maximum frequency of 20 Hz." Are the detector and electronics signal chain sufficiently fast to resolve changes in water vapor at 20 Hz?*

Yes, the provided maximum frequency is 20 Hz. The internal data sampling and acquisition is fast enough. To avoid confusion, we changed the sentence to:

"The data sampled internally at 150 Hz frequency is processed and provided at a maximum frequency of 20 Hz."

*4.2 Page 5, line 9: what is the response time of the Rosemount EL102 sensor? It is only characterized here as a "fast response time." How fast?* We changed the text to:// "Rosemount EL102 sensor with a fast response time (100 Hz)"

*4.3 Page 5, lines 25-30: How can you carry out successful fast measurements with the closed-path LICOR if there is a 250-millisecond calculated delay? Have you tested the delay? What is the residence time in the sample cell?*

The delay is just a temporal offset, which can be corrected. The residence time in the sample cell depends on the air flow speed, and is taken into account in Sect. 2.2.

*4.4 Page 10, line 23 and Page 11, line 25: the authors state the "Generally a temporal resolution of 20 Hz is sufficient for humidity flux calculations." It is not clearly explained how the authors came to this conclusion. Is there a reference that can be cited as evidence for this?*

We included in Section 4.2 the following text:

"It can be concluded from Fig.10 and Fig.12 that the fluxes for frequencies exceeding 1 Hz are negligible for these specific flight conditions. Therefore, the sampling frequency of 20 Hz is sufficient for airborne turbulent humidity fluxes."

*Furthermore, it is unclear from this manuscript whether the LICOR has an actual temporal resolution of 20 Hz (when the sampling delays and internal processing are included).*

The delays are constant temporal offsets, which do not influence the capability of the sensor to provide data at 20 Hz resolution.

*5) Flight figures 3 and 6 are hard to read: please consider larger font text*

We enlarged the text in the figures.

**References**

Buchholz, B., Kühnreich, B., Smit, H.G.J., and Ebert, V.: Validation of an extractive, airborne, compact TDL spectrometer for atmospheric humidity sensing by blind intercomparison, Appl. Phys. B, 110, 249-262, 2013.

5    Buchholz, B., Böse, N., and Ebert, V.: Absolute validation of a diode laser hygrometer via intercomparison with the German national primary water vapor standard, Appl. Phys. B, 116, 4, 883-899, 2014.

Buchholz, B., Afchine, A., Klein, A., Schiller, C., Krämer, M., and Ebert, V.: HAI, a new airborne, absolute, twin dual-channel, multi-phase TDLAS-hygrometer: background, design, setup, and first flight data, Atmos. Meas. Tech., 9, 1-23, 2016.

Herman, R.L., Ray, E.A., Rosenlof, K.H., Bedka, K.M., Schwartz, M.J., Read, W.G., Troy, R.F., Chin, K., Christensen, L.E., Fu, D., Stachnik,

10    R.A., Bui, T.P., Dean-Day, J.M.: Enhanced stratospheric water vapor over the summertime continental United States and the role of overshooting convection, Atmos. Chem. Phys., 17, 6113-6124, 2017.

May, R.D.: Open-path, near-infrared tunable diode laser spectrometer for atmospheric measurements of $H_2O$, J., Geophys. Res., 103, D15, 19161-19172, 1998.

Zondlo, M.A., Paige, M.E., Massick, S.M., and Silver, J.A., Vertical cavity laser hygrometer for the National Science Foundation Gulfstream-V aircraft, J. Geophys. Res., 115, D20309, doi:10.1029/2010JD014445, 14 pp., 2010.

---

## Editor Decision (ED1)

Dear Astrid Lampert and co-authors,

thanks for your detailed replies to the reviewer comments and providing a revised manuscript, which is much improved.

Please find below remarks on your revised manuscript, following up on the reviewers' comments, where I find that further clarification is needed and/or would increase the quality of your manuscript. (page and line numbers refer to the revised manuscript)

Abstract:

Maybe add a sentence that explains your motivation. Why are you conducting this study, why is this comparison important?

Introduction:

**p1, l19:** Maybe add an example (e.g. "as towers"): " ... and measurements at fixed locations, as towers, providing higher vertical and temporal resolution." (In accordance with reviewer 1 – it's a point measurement - I would omit the part "but representative only for a small area")

**p1, l21ff**: This small section about the turbulence and fluxes stands a bit out of context here. However, as it talks about essential properties for your study, I suggest better integration it in the text and extension of the content.

**p2, l4-6:** I feel that your explanation to the reviewer regarding the relevance of cloud chamber studies to your study does a better job than the sentence you write there. Maybe rephrase similar to:

"Since measuring atmospheric water vapour precisely is difficult, the uncertainties of atmospheric water vapour measurements are high.  Even with the best systems under well controlled conditions in the laboratory, there are large discrepancies between different measurement systems, e.g. intercomparison measurements of different hygrometers probing the same air simultaneously revealed discrepancies between different measurement systems of around 10 %."

Please also include a reference for your last statement ("discrepancies between different measurement systems of around 10 %").

**p2, l16:** ..., which is used..." (is instead of are)

**p3, l9-11:** Maybe better: "However, with the end of the life time of the radiation sources (glow discharge lamps) and difficulties in replacing them, other humidity sensors become more important, and a variety of fast-response sensors is now available."

However, in the following you only mention two more sensors (one not being applied for atmospheric research, the other specific to one research aircraft), this does not sound like a "variety of fast-response sensors"? Please clarify!

Section 1.2/1.3:

I find it somewhat confusing that you refer to molecular absorption (and their deficiencies compared to atomic absorption) in Section 1.2 before introducing molecular absorption in Section 1.3. Either swap these sections or refer to atomic absorption from section 1.3 instead.

**p3, l11:** "A similar system is the Krypton hygrometer KH20" similar to what? State!

**p3, l13:** "It is, however, not broadly present in airborne applications." Maybe drag some text of your reply to the reviewer into the manuscript:

"It is, however, mostly used for ground-based measurements. Furthermore, the instrument is very sensitive to path length, and calibration is difficult even for the ground-based applications (Foken and Falke, 2010)."

**p3, l19:** "...are now easily available..." Do you mean "readily" available?

**p4, l8ff:** I find this little section deserves more attention, as it points out the importance of your study. I recommend strengthening this part, stating your motivation more clearly (cf your reply to reviewer comment

> "*I have a concern that the authors and other research groups are using LICOR sensors in an environment that the manufacturer does not recommend.*
>
> We understand that the manufacturer does not sell the system for airborne applications, as they are aware that vibrations may hamper the data. For that reason, we consider it even more important to figure out the limitations of the sensor in terms of vibrations.)"

Maybe add something along the line: "...its airborne applications will very likely increase. Therefore, knowing the limitations of the LICOR sensors with respect to vibrations is important, and one of the main aims of this study."

**p4, l21:** "The spectroscopic sensors are experimental systems and not commercially available yet." Remove "yet", these systems probably remain experimental and will never be commercial.

**p4, l25:** "... for the typical flight altitude of few 100m and airspeed..." – "of a few 100m"

**p5, l12/13:** Regarding your reply to reviewer comment about the "delay time":

> "*How can you carry out successful fast measurements with the closed-path LICOR if there is a 250-millisecond calculated delay? Have you tested the delay? What is the residence time in the sample cell?*
>
> The delay is just a temporal offset, which can be corrected. The residence time in the sample cell depends on the air flow speed, and is taken into account in Sect. 2.2."

also:

> "*Furthermore, it is unclear from this manuscript whether the LICOR has an actual temporal resolution of 20 Hz (when the sampling delays and internal processing are included).*
>
> The delays are constant temporal offsets, which do not influence the capability of the sensor to provide data at 20 Hz resolution.)"

Maybe better say "temporal offset" "or "time shift" instead of "delay time" or "time delay" to avoid confusion here and in other places?

**p5, l19/20:** "...a slow, but highly accurate Rosemount DB102 temperature sensor..." Please specify slow and highly accurate.

**p5, l20:** "... and a a fast response (100Hz) Rosemount EL102 sensor."

Remove one "a", and add what quantity the Rosemount EL102 sensor is measuring.

**p5, l22/23:** Regarding the reviewer comment about Humicap: If you do not use data from this instrument, I suggest not mentioning it, or at least say "(not operational in this study)" instead of "(not used for this study)".

**p6, l3-9:** I agree with the reviewer comment about the time resolution discussion being split up in too many places, but also understand your reasoning. The discussion here in Section 2.2 could be moved into and combined with Section 3.2?!

**former p6, l20ff:** reply to reviewer comment:

> "We added in the text:

> "These small sub-legs were chosen with different but homogeneous surface conditions and different but constant flight altitudes to compare if there are systematic differences in the parameters like the vibration level."

> "

I could not find the added text in the revised manuscript!

**p7, l30:** Fig. 9 referenced before Fig. 4. Please number figures in accordance with their mentioning in the text.

**p8, l2:** "The overall aim of the Helipod measurements was to study greenhouse gas emissions on a scale of up to 100 km..."

What does the scale of up to 100km refer to?

**former p10, l12:** reply to reviewer comment:

> *"... covariance of the vertical wind speed and the humidity values from the different sensors..."*

> We changed as suggested.

The change got lost in the revised manuscript?!

Figures:

**General:** I recommend using 90° turned y-axis labels in all figures (not all labels turned in figures 3, 5, 9, and 10)

**Fig. 5:** "For the spectral analysis, the part of the data shaded in grey were used, excluding segments indicated in the last plot." You mean "... lowermost panel" or "...panel e" (not "last plot")?

Also change "...the part of the data... was used..."

**Fig. 6:** Does it refer to the shaded area in Fig.3? If yes, please state explicitly; if no, please indicate the relevant section in Fig. 3 (e.g. with a bar on top or bottom of the figure if you wish to add no further shadings).

**Fig. 9:** Some labels and y-axis annotations of the left plot overlap: "180°" phase and "0" coherence, "phase" and "0°" phase.

---

## Author Response (AR2)

**Answers to the Editor: Comparison of the fast Lyman-Alpha and LICOR hygrometers for measuring airborne turbulent fluctuations, revised version**

Astrid Lampert[1], Jörg Hartmann[2], Falk Pätzold[1], Lennart Lobitz[1], Peter Hecker[1], Katrin Kohnert[3], Eric Larmanou[3,4], Andrei Serafimovich[3], and Torsten Sachs[1,3]

[1]Institute of Flight Guidance, TU Braunschweig, Hermann-Blenk-Str. 27, 38108 Braunschweig, Germany
[2]Alfred Wegener Institute for Polar and Marine Research, Bussestr. 24, 27570 Bremerhaven, Germany
[3]GFZ German Research Centre for Geosciences, Telegrafenberg, 14473 Potsdam, Germany
[4]Swedish University of Agricultural Sciences, Umeå, Sweden

*Correspondence to:* Astrid Lampert (Astrid.Lampert@tu-braunschweig.de)

The authors would like to thank the editor for the questions and remarks which helped to further improve the manuscript. In the following, each comment of the editor (in italic) is answered separately. The answers are provided in normal style, and the changed text of the manuscript is given in quotation marks. Further, changes to the manuscript are highlighted in the latexdiff version.

*Abstract:*

*Maybe add a sentence that explains your motivation. Why are you conducting this study, why is this comparison important?*

We changed the first sentence to: "To investigate if the LICOR humidity sensor can be used as a replacement of the Lyman-Alpha sensor for airborne applications, the measurement data of the Lyman-Alpha and several LICOR sensors are analysed in direct intercomparison flights on different airborne platforms."

*Introduction:*

*p1, l19: Maybe add an example (e.g. "as towers"): " ... and measurements at fixed locations, as towers, providing higher vertical and temporal resolution." (In accordance with reviewer 1 – it's a point measurement - I would omit the part "but representative only for a small area")*

We changes as suggested.

*p1, l21ff: This small section about the turbulence and fluxes stands a bit out of context here. However, as it talks about essential properties for your study, I suggest better integration it in the text and extension of the content.*

We expanded the text to "The most effective way for moisture transport from the surface to the atmosphere is turbulence. Turbulent fluxes are commonly determined with the eddy-covariance method. This technique requires accurate and high resolution measurements of the fluctuations of the vertical component of wind speed and humidity." Further, we put the text in a more suitable place in the introduction, before introducing the requirements for airborne measurements of humidity.

*p2, l4-6: I feel that your explanation to the reviewer regarding the relevance of cloud chamber studies to your study does a better job than the sentence you write there. Maybe rephrase similar to: "Since measuring atmospheric water vapour pre-*

*cisely is difficult, the uncertainties of atmospheric water vapour measurements are high. Even with the best systems under well controlled conditions in the laboratory, there are large discrepancies between different measurement systems, e.g. intercomparison measurements of different hygrometers probing the same air simultaneously revealed discrepancies between different measurement systems of around 10 %."*

We changed as suggested.

*Please also include a reference for your last statement ("discrepancies between different measurement systems of around 10 %").*

It is the same as for the next sentence. We clarified in the text.

*p2, l16: ..., which is used..." (is instead of are)*

We corrected this.

*p3, l9-11: Maybe better: "However, with the end of the life time of the radiation sources (glow discharge lamps) and difficulties in replacing them, other humidity sensors become more important, and a variety of fast-response sensors is now available."*

We changed as suggested.

*However, in the following you only mention two more sensors (one not being applied for atmospheric research, the other specific to one research aircraft), this does not sound like a "variety of fast-response sensors"? Please clarify!* We added in the text: "e.g. the following two sensors"

*Section 1.2/1.3:*

*I find it somewhat confusing that you refer to molecular absorption (and their deficiencies compared to atomic absorption) in Section 1.2 before introducing molecular absorption in Section 1.3. Either swap these sections or refer to atomic absorption from section 1.3 instead.*

Agree - we deleted the comparison with molecular absorption in Sect. 1.2, and compare the required length of the measurement cell in Sect. 1.3, after having introduced molucular absorption.

*p3, l11: "A similar system is the Krypton hygrometer KH20" similar to what? State!*

We changed the text to: "A system similar to the Lyman-Alpha is the Krypton hygrometer KH20"

*p3, l13: "It is, however, not broadly present in airborne applications." Maybe drag some text of your reply to the reviewer into the manuscript: "It is, however, mostly used for ground-based measurements. Furthermore, the instrument is very sensitive to path length, and calibration is difficult even for the ground-based applications (Foken and Falke, 2010)."*

We changed as suggested.

*p3, l19: "...are now easily available..." Do you mean "readily" available?*

Yes, we changed that.

*p4, l8ff: I find this little section deserves more attention, as it points out the importance of your study. I recommend strengthening this part, stating your motivation more clearly (cf your reply to reviewer comment "I have a concern that the authors and other research groups are using LICOR sensors in an environment that the manufacturer does not recommend. We understand that the manufacturer does not sell the system for airborne applications, as they are aware that vibrations may hamper the*

*data. For that reason, we consider it even more important to figure out the limitations of the sensor in terms of vibrations.)"*
*Maybe add something along the line: "...its airborne applications will very likely increase. Therefore, knowing the limitations*
*of the LICOR sensors with respect to vibrations is important, and one of the main aims of this study."*
We changed as suggested.

5 *p4, l21: "The spectroscopic sensors are experimental systems and not commercially available yet." Remove "yet", these*
*systems probably remain experimental and will never be commercial.*
We removed "yet".

*p4, l25: "... for the typical flight altitude of few 100m and airspeed..." – "of a few 100m"*
We changed as suggested.

10 *p5, l12/13: Regarding your reply to reviewer comment about the "delay time": "How can you carry out successful fast*
*measurements with the closed-path LICOR if there is a 250-millisecond calculated delay? Have you tested the delay? What is*
*the residence time in the sample cell? The delay is just a temporal offset, which can be corrected. The residence time in the*
*sample cell depends on the air flow speed, and is taken into account in Sect. 2.2." also: "Furthermore, it is unclear from this*
*manuscript whether the LICOR has an actual temporal resolution of 20 Hz (when the sampling delays and internal processing*
15 *are included). The delays are constant temporal offsets, which do not influence the capability of the sensor to provide data*
*at 20 Hz resolution.)" Maybe better say "temporal offset" "or "time shift" instead of "delay time" or "time delay" to avoid*
*confusion here and in other places?*
We replaced "time delay" by "time shift" throughout the text.

*p5, l19/20: "...a slow, but highly accurate Rosemount DB102 temperature sensor..." Please specify slow and highly accurate.*
20 We changed the text to: "a long-time stable Rosemount DB102 temperature sensor with slow response time of around 1 s"

*p5, l20: "... and a a fast response (100Hz) Rosemount EL102 sensor." Remove one "a", and add what quantity the Rosemount*
*EL102 sensor is measuring.*
We added "temperature" sensor.

*p5, l22/23: Regarding the reviewer comment about Humicap: If you do not use data from this instrument, I suggest not*
25 *mentioning it, or at least say "(not operational in this study)" instead of "(not used for this study)".*
This comment refers to the dew point mirror, not the Humicap. For the Do128, we changed the text to "not operational for this
study"

*p6, l3-9: I agree with the reviewer comment about the time resolution discussion being split up in too many places, but also*
*understand your reasoning. The discussion here in Section 2.2 could be moved into and combined with Section 3.2?!*
30 As a compromise, we changed the order of the Sections. Now we have the Section Do-128 Instrumentation, then the Section
Do-128 time synchronisation, then the same for Helipod.

*former p6, l20ff: reply to reviewer comment: "We added in the text: "These small sub-legs were chosen with different but*
*homogeneous surface conditions and different but constant flight altitudes to compare if there are systematic differences in the*
*parameters like the vibration level." " I could not find the added text in the revised manuscript!*
35 Sorry, this sentence got lost. Now we added in the text: "Such small sub-legs were chosen with different but homogeneous

surface conditions and different but constant flight altitudes to compare if there are systematic differences in the parameters like the vibration level."

*p7, l30: Fig. 9 referenced before Fig. 4. Please number figures in accordance with their mentioning in the text.*

We checked again the order of the figures according to the text.

*p8, l2: "The overall aim of the Helipod measurements was to study greenhouse gas emissions on a scale of up to 100 km..." What does the scale of up to 100km refer to?*

We re-phrased the sentence: "The overall aim of the Helipod measurements was to study greenhouse gas emissions on a climatically relevant sub-regional scale of up to 100 km to investigate the spatial variability, and to analyse how representative the continuous emission measurements on local scales are on this larger scale."

*former p10, l12: reply to reviewer comment: "... covariance of the vertical wind speed and the humidity values from the different sensors..." We changed as suggested. The change got lost in the revised manuscript?!*

Thank you for the hint! We corrected this as originally planned.

Figures:

*General: I recommend using 90 degree turned y-axis labels in all figures (not all labels turned in figures 3, 5, 9, and 10)*

For the time series with multiple panels (Fig. 3 and 5), we prefer to use horizontal y-axis labels, as there is not enough space otherwise. We changed the orientation of the labels for all sub-plots in Fig. 9. Also for Fig. 10, the y-axis is only labelled with a short formula, which is much easier to understand if it is not rotated.

*Fig. 5: "For the spectral analysis, the part of the data shaded in grey were used, excluding segments indicated in the last plot." You mean "... lowermost panel" or "...panel e" (not "last plot")?*

We changed this.

*Also change "...the part of the data... was used..."*

We corrected this.

*Fig. 6: Does it refer to the shaded area in Fig.3? If yes, please state explicitly; if no, please indicate the relevant section in Fig. 3 (e.g. with a bar on top or bottom of the figure if you wish to add no further shadings).*

We indicated the part of the flight used for the vibration analyses in the time series, and added a reference to the time series in the caption of the figures showing the vibration time series and spectra.

*Fig. 9: Some labels and y-axis annotations of the left plot overlap: "180 degree" phase and "0" coherence, "phase" and "0 degree" phase.*

We changed that.